# The art of using t-SNE for single-cell transcriptomics

Dmitry Kobak [1]* & Philipp Berens [1,2,3,4]*

Single-cell transcriptomics yields ever growing data sets containing RNA expression levels for thousands of genes from up to millions of cells. Common data analysis pipelines include a dimensionality reduction step for visualising the data in two dimensions, most frequently performed using t-distributed stochastic neighbour embedding (t-SNE). It excels at revealing local structure in high-dimensional data, but naive applications often suffer from severe shortcomings, e.g. the global structure of the data is not represented accurately. Here we describe how to circumvent such pitfalls, and develop a protocol for creating more faithful t-SNE visualisations. It includes PCA initialisation, a high learning rate, and multi-scale similarity kernels; for very large data sets, we additionally use exaggeration and downsampling-based initialisation. We use published single-cell RNA-seq data sets to demonstrate that this protocol yields superior results compared to the naive application of t-SNE.

[1] Institute for Ophthalmic Research, University of Tübingen, Tübingen, Germany. [2] Bernstein Center for Computational Neuroscience, University of Tübingen, Tübingen, Germany. [3] Center for Integrative Neuroscience, University of Tübingen, Tübingen, Germany. [4] Institute for Bioinformatics and Medical Informatics, University of Tübingen, Tübingen, Germany. *email: dmitry.kobak@uni-tuebingen.de; philipp.berens@uni-tuebingen.de

Recent years have seen a rapid growth of interest in single-cell RNA sequencing (scRNA-seq), or single-cell transcriptomics[1,2]. Through improved experimental techniques it has become possible to obtain gene expression data from thousands or even millions of cells[3–8]. Computational analysis of such data sets often entails unsupervised, exploratory steps including dimensionality reduction for visualisation. To this end, many studies today are using t-distributed stochastic neighbour embedding, or t-SNE[9].

This technique maps a set of high-dimensional points to two dimensions, such that ideally, close neighbours remain close and distant points remain distant. Informally, the algorithm places all points on the 2D plane, initially at random positions, and lets them interact as if they were physical particles. The interaction is governed by two laws: first, all points are repelled from each other; second, each point is attracted to its nearest neighbours (see Methods for a mathematical description). The most important parameter of t-SNE, called perplexity, controls the width of the Gaussian kernel used to compute similarities between points and effectively governs how many of its nearest neighbours each point is attracted to. The default value of perplexity in existing implementations is 30 or 50 and the common wisdom is that "the performance of t-SNE is fairly robust to changes in the perplexity"[9].

When applied to high-dimensional but well-clustered data, t-SNE tends to produce a visualisation with distinctly isolated clusters, which often are in good agreement with the clusters found by a dedicated clustering algorithm. This attractive property as well as the lack of serious competitors until very recently[10,11] made t-SNE the de facto standard for visual exploration of scRNA-seq data. At the same time, t-SNE has well known, but frequently overlooked weaknesses[12]. Most importantly, it often fails to preserve the global geometry of the data. This means that the relative position of clusters on the t-SNE plot is almost arbitrary and depends on random initialisation more than on anything else. While this may not be a problem in some situations, scRNA-seq data sets often exhibit biologically meaningful hierarchical structure, e.g. encompass several very different cell classes, each further divided into various types. Typical t-SNE plots do not capture such global structure, yielding a suboptimal and potentially misleading visualisation. In our experience, the larger the data set, the more severe this problem becomes. Other notable challenges include performing t-SNE visualisations for very large data sets (e.g. a million of cells or more), or mapping cells collected in follow-up experiments onto an existing t-SNE visualisation.

Here we explain how to achieve improved t-SNE visualisations that preserve the global geometry of the data. Our method relies on providing PCA initialisation, employing multi-scale similarities[13,14], increasing the learning rate[15], and for very large data sets, additionally using the so called exaggeration and downsampling-based initialisation. To demonstrate these techniques we use several full-length and UMI-based data sets with up to two million cells (Table 1). We use FIt-SNE[16], a recently developed fast t-SNE implementation, for all experiments.

In many challenging cases, our t-SNE pipeline yields visualisations that are better than the state of the art. We discuss its advantages and disadvantages compared to UMAP[10], a recent dimensionality reduction method that is gaining popularity in the scRNA-seq community[11]. We also describe how to position new cells on an existing t-SNE reference atlas and how to visualise multiple related data sets side by side in a consistent fashion. We focus on single-cell transcriptomics but our recommendations are more generally applicable to any data set that has hierarchical organisation, which is often the case e.g. in single-cell flow or mass cytometry[17,18], whole-genome sequencing[19,20], as well as outside of biology[21].

## Results

**Preserving global geometry with t-SNE.** To illustrate that the default t-SNE tends to misrepresent the global geometry, we first consider a toy example (Fig. 1). This synthetic data set consists of points sampled from fifteen 50-dimensional spherical Gaussian distributions, grouped into three distinct and non-overlapping classes. The data are generated such that the types within two classes ($n = 100$ and $n = 1000$ per type, respectively) do not overlap, and the types within the third class ($n = 2000$ per type) are partially overlapping. As a result, this data set exhibits hierarchical structure, typical for scRNA-seq data.

Two classical methods to visualise high-dimensional data are multidimensional scaling (MDS) and principal component analysis (PCA). MDS is difficult to compute with a large number of points (here $n = 15,500$), but can be easily applied to class means ($n = 15$), clearly showing the three distinct classes (Fig. 1a). PCA can be applied to the whole data set and demonstrates the same large-scale structure of the data (Fig. 1b), but no within-class structure can be seen in the first two PCs. In contrast, t-SNE clearly shows all 15 types, correctly displaying ten of them as fully isolated and five as partially overlapping (Fig. 1c). However, the isolated types end up arbitrarily placed, with their positions mostly depending on the random seed used for initialisation.

In order to quantify numerically the quality, or faithfulness, of a given embedding, we used three different metrics:

KNN   The fraction of $k$-nearest neighbours in the original high-dimensional data that are preserved as $k$-nearest neighbours in the embedding[22]. We used $k = 10$ and computed the average across all $n$ points. KNN quantifies preservation of the local, or microscopic structure.

KNC   The fraction of $k$-nearest class means in the original data that are preserved as $k$-nearest class means in the embedding. This is computed for class means only and averaged across all classes. For the synthetic data set we used $k = 4$, and for the real data sets analysed below we used $k = 10$. KNC quantifies preservation of the mesoscopic structure.

CPD   Spearman correlation between pairwise distances in the high-dimensional space and in the embedding[11]. Computed across all 499,500 pairs among 1000 randomly chosen points. CPD quantifies preservation of the global, or macroscropic structure.

Applying these metrics to the PCA and t-SNE embeddings (Fig. 1b, c) shows that t-SNE is much better than PCA in preserving the local structure (KNN 0.13 vs. 0.00) but much worse in preserving the global structure (KNC 0.23 vs. 1.00 and CPD 0.51 vs. 0.85). Our recipe for a more faithful t-SNE visualisation is based on three ideas that have been previously suggested in various contexts: multi-scale similarities[13,14], PCA initialisation, and increased learning rate[15].

Fig. 1c used perplexity 30, which is the default value in most t-SNE implementations. Much larger values can yield qualitatively different outcomes. As large perplexity yields longer-ranging attractive forces during t-SNE optimisation, the visualisation loses some fine detail but pulls larger structures together. As a simple rule of thumb, we take 1% of the sample size as a large perplexity for any given data set; this corresponds to perplexity 155 for our simulated data and results in five small clusters belonging to the same class attracted together (Fig. 1d). Our metrics confirmed that, compared to the standard perplexity value, the local structure (KNN) deteriorates but the global structure (KNC and CPD) improves. A multi-scale approach, using multiple perplexity values at the same time, has been previously suggested to preserve both local and global structure[13,14]. We adopt this

**Table 1 Data sets used in this study, listed in the order of appearance in the text. In all cases, we relied on quality control and clustering performed in the original publications. For the 10x Genomics data set we used cluster labels from ref. [23].**

| Data set name | Protocol | Organism and tissue | No. of cells | No. of classes |
|---|---|---|---|---|
| Tasic et al.[3] | Smart-seq2 | adult mouse cortex | 23,822 | 133 |
| Macosko et al.[24] | Drop-seq | mouse retina | 44,808 | 39 |
| Shekhar et al.[25] | Drop-seq | mouse retina | 27,499 | 26 |
| Harris et al.[26] | 10x Chromium | mouse hippocampus | 3663 | 49 |
| Cadwell et al.[27] | Smart-seq2 (Patch-seq) | adult mouse cortex | 46 | 2 |
| Tasic et al.[31] | SMARTer | adult mouse cortex | 1679 | 49 |
| 10x Genomics | 10x Chromium | mouse embryonic brain | 1,306,127 | 39 |
| Cao et al.[8] | sci-RNA-seq3 | mouse embryo | 2,058,652 | 38 |

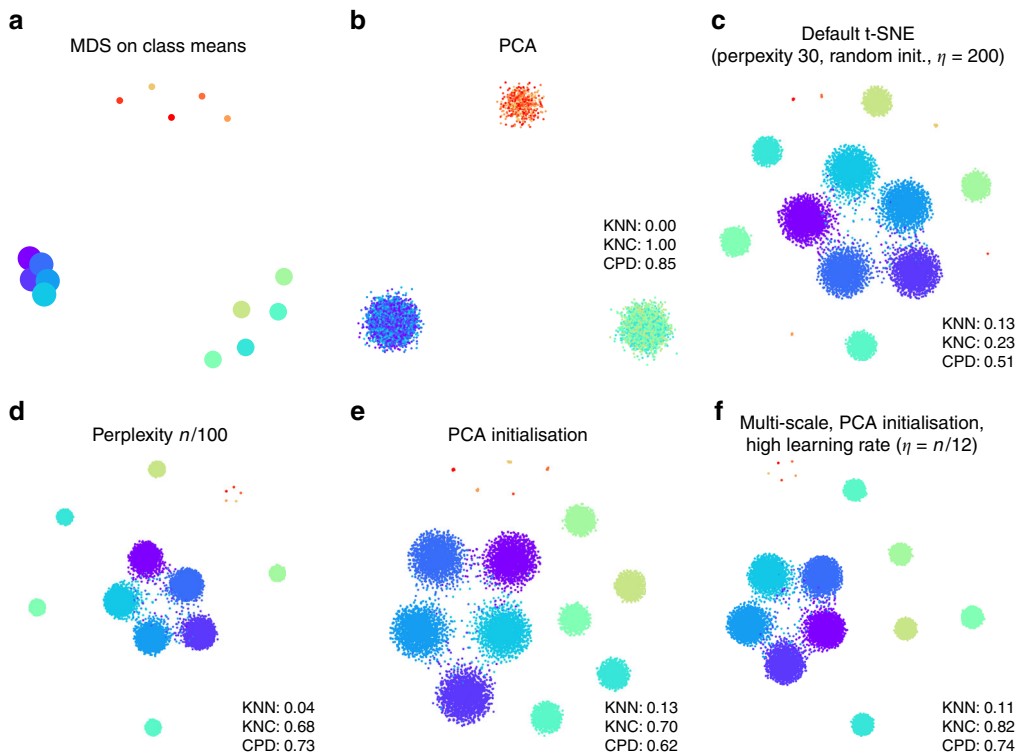

**Fig. 1** Synthetic data set. The points were sampled from a mixture of fifteen 50-dimensional Gaussian distributions. Total sample size $n = 15,500$.
**a** Multidimensional scaling of 15 class means. Point sizes are proportional to the number of points per class. **b** The first two principal components of the data. Point colours denote class membership. KNN: 10-nearest neighbour preservation, KNC: 4-nearest classes preservation, CPD: Spearman correlation between pairwise distances. **c** Default t-SNE with perplexity 30, random initialisation, and learning rate 200. **d** T-SNE with perplexity $n/100 = 155$. **e** T-SNE with PCA initialisation. **f** T-SNE with multi-scale similarities (perplexity combination of 30 and $n/100 = 155$), PCA initialisation, and learning rate $n/12 \approx 1300$.

approach in our final pipeline and, whenever $n/100 \gg 30$, combine perplexity 30 with the large perplexity $n/100$ (see below; separate evaluation not shown here).

Another approach to preserve global structure is to use an informative initialisation, e.g. the first two PCs (after appropriate scaling, see Methods). This injects the global structure into the t-SNE embedding which is then preserved during the course of t-SNE optimisation while the algorithm optimises the fine structure (Fig. 1e). Indeed, KNN did not depend on initialisation, but both KNC and CPD markedly improved when using PCA initialisation. PCA initialisation is also convenient because it makes the t-SNE outcome reproducible and not dependent on a random seed.

The third ingredient in our t-SNE protocol is to increase the learning rate. The default learning rate in most t-SNE implementations is $\eta = 200$ which is not enough for large data

sets and can lead to poor convergence and/or convergence to a suboptimal local minimum[15]. A recent Python library for scRNA-seq analysis, scanpy, increased the default learning rate to 1000[23], whereas ref. [15] suggested to use $\eta = n/12$. We adopt the latter suggestion and use $\eta = n/12$ whenever it is above 200. This does not have a major influence on our synthetic data set (because its sample size is not large enough for this to matter), but will be important later on.

Putting all three modifications together, we obtain the visualisation shown in Fig. 1f. The quantitative evaluation confirmed that in terms of the mesoscopic/macroscopic structure, our suggested pipeline strongly outperformed the default t-SNE and was better than large perplexity or PCA initialisation on their own. At the same time, in terms of the miscroscopic structure, it achieved a compromise between the small and the large perplexities.

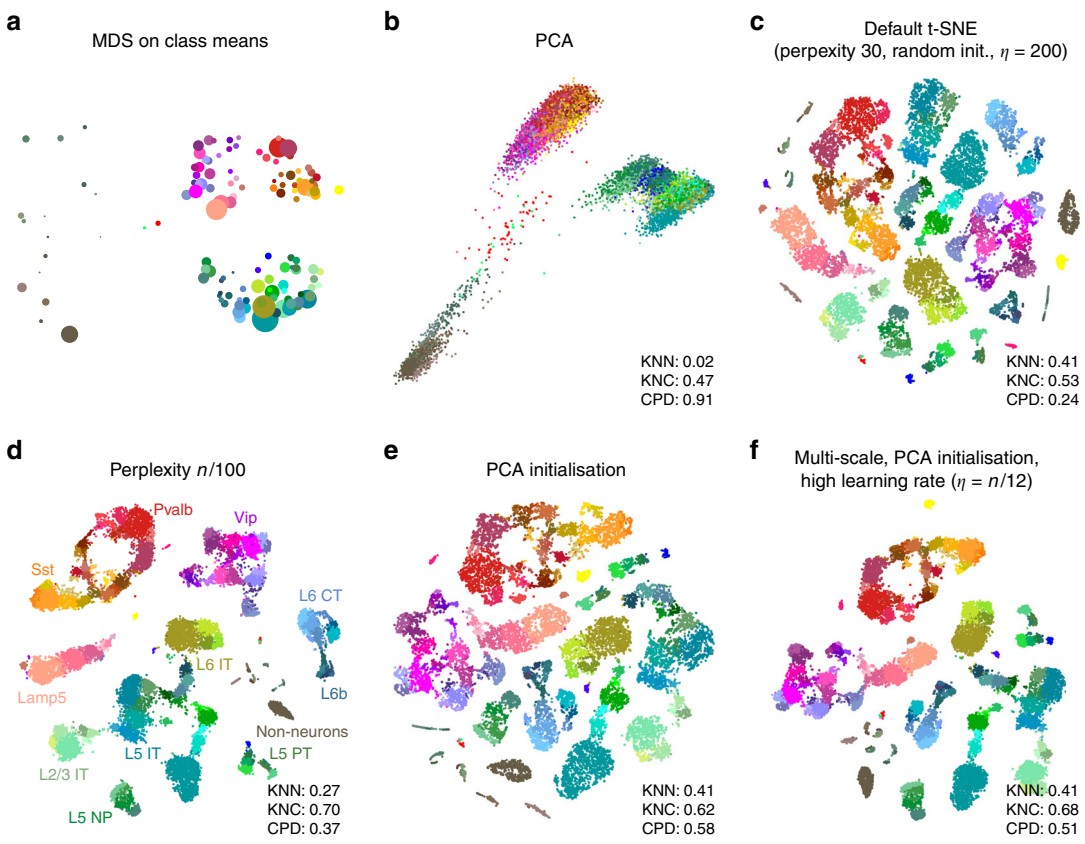

**Fig. 2** Tasic et al. data set. Sample size $n = 23,822$. Cluster assignments and cluster colours are taken from the original publication[3]. Warm colours correspond to inhibitory neurons, cold colours correspond to excitatory neurons, brown/grey colours correspond to non-neural cells. **a** MDS on class means ($n = 133$). Point sizes are proportional to the number of points per class. **b** The first two principal components of the data. KNN: 10-nearest neighbour preservation, KNC: 10-nearest classes preservation, CPD: Spearman correlation between pairwise distances. **c** Default t-SNE with perplexity 30, random initialisation, and learning rate 200. **d** T-SNE with perplexity $n/100 = 238$. Labels denote large groups of clusters. **e** T-SNE with PCA initialisation. **f** T-SNE with multi-scale similarities (perplexity combination of 30 and $n/100 = 238$, PCA initialisation, and learning rate $n/12 \approx 2000$.

**Faithful t-SNE of transcriptomic data sets**. To demonstrate these ideas on a real-life data set, we chose to focus on the data set from Tasic et al.[3]. It encompasses 23,822 cells from adult mouse cortex, split by the authors into 133 clusters with strong hierarchical organisation. Here and below we used a standard preprocessing pipeline consisting of sequencing depth normalisation, feature selection, log-transformation, and reducing the dimensionality to 50 PCs (see Methods).

In the Tasic et al. data, three well-separated groups of clusters are apparent in the MDS (Fig. 2a) and PCA (Fig. 2b) plots, corresponding to excitatory neurons (cold colours), inhibitory neurons (warm colours), and non-neural cells such as astrocytes or microglia (grey/brown colours). Performing PCA on these three data subsets separately (Supplementary Fig. 1) reveals further structure inside each of them: e.g. inhibitory neurons are well separated into two groups, *Pvalb/SSt*-expressing (red/yellow) and *Vip/Lamp5*-expressing (purple/salmon), as can also be seen in Fig. 2a. This demonstrates the hierarchical organisation of the data.

This global structure is missing from a standard t-SNE visualisation (Fig. 2c): excitatory neurons, inhibitory neurons, and non-neural cells are all split into multiple islands that are shuffled among each other. For example, the group of purple clusters (*Vip* interneurons) is separated from a group of salmon clusters (a closely related group of *Lamp5* interneurons) by some excitatory clusters, misrepresenting the hierarchy of cell types. This outcome is not a strawman: the original paper[3] features a t-SNE figure qualitatively very similar to our visualisation.

Perplexity values in the common range (e.g. 20, 50, 80) yield similar results, confirming that t-SNE is not very sensitive to the exact value of perplexity.

In contrast, setting perplexity to 1% of the sample size, in this case to 238, pulls together large groups of related types, improving the global structure (KNC and CPD increase), at the expense of losing some of the fine structure (KNN decreases, Fig. 2d). PCA initialisation with default perplexity also improves the global structure (KNC and CPD increase, compared to the default t-SNE, Fig. 2e). Finally, our suggested pipeline with multi-scale similarities (perplexity combination of 30 and $n/100 = 238$), PCA initialisation, and learning rate $n/12 \approx 2000$ yields an embedding with high values of all three metrics (Fig. 2f). Compared to the default parameters, these settings slowed down FIt-SNE from ~30 s to ~2 m, which we still find to be an acceptable runtime.

It is instructive to study systematically how the choice of parameters influences the embedding quality (Fig. 3). We found that the learning rate only influences KNN: the higher the learning rate, the better preserved is the local structure, until is saturates at around $n/10$ (Fig. 3a), in agreement with the results of ref. [15]. The other two metrics, KNC and CPD, are not affected by the learning rate (Fig. 3c, e). The perplexity controls the trade-off between KNN and KNC: the higher the perplexity combined with 30, the worse is the microscropic structure (Fig. 3b) but the better is the mesoscopic structure (Fig. 3d). Our choice of $n/100$ provides a reasonable compromise. Finally, the PCA initialisation strongly improves the macroscopic structure as measured by the

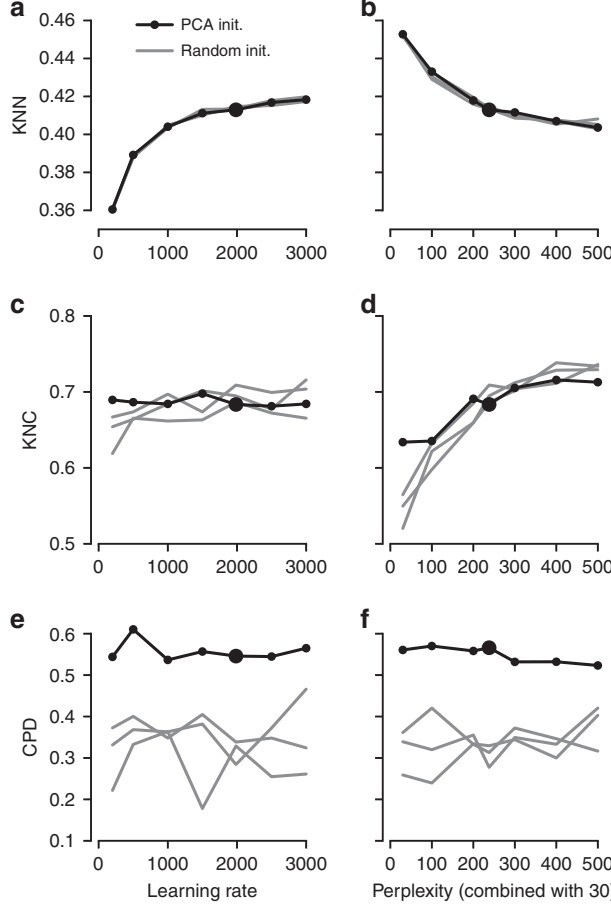

**Fig. 3** The influence of parameter values on embedding quality. All panels show quality assessments of various t-SNE embeddings of the Tasic et al. data set. **a** 10-nearest neighbours preservation (KNN) as a function of learning rate. Black line shows PCA initialisation, grey lines show random initialisations with three different random seeds. Large black dot denotes our preferred parameter values. Perplexity combination of 30 and $n/100$. **b** 10-nearest neighbours preservation as a function of perplexity used in combination with perplexity 30. Learning rate $n/12$. **c–d** The same for 10-nearest classes preservation (KNC). **e–f** The same for Spearman correlation between pairwise distances (CPD).

CPD (Fig. 3e, f), while the other two parameters have little influence on it.

To demonstrate that our approach is equally well applicable to UMI-based transcriptomic data, we considered three further data sets. First, we analysed a $n = 44,808$ mouse retina data set from ref. [24]. Our t-SNE result preserved much of the global geometry (Fig. 4a): e.g. multiple amacrine cell clusters (green), bipolar cell clusters (blue), and non-neural clusters (magenta) were placed close together. The t-SNE analysis performed by the authors in the original publication relied on downsampling and had a worse representation of the cluster hierarchy.

Second, we analysed a $n = 27,499$ data set from ref. [25] that sequenced cells from mouse retina targeting bipolar neurons. Here again, our t-SNE result (Fig. 4b) is consistent with the global structure of the data: for example, OFF bipolar cells (types 1–4, warm colours) and ON bipolar cells (types 5–9, cold colours) are located close to each other, and four subtypes of type 5 are also close together. This was not true for the t-SNE shown in the original publication. This data set shows one limitation of our method: the data contain several very distinct but very rare

clusters and those appear in the middle of the embedding, instead of being placed far out on the periphery (see Discussion).

Finally, we analysed a $n = 3663$ data set of hippocampal interneurons from ref. [26]. The original publication introduced a novel clustering and feature selection method based on the negative binomial distribution, and used a modified negative binomial t-SNE procedure. Our t-SNE visualisation (Fig. 4c) did not use any of that but nevertheless led to an embedding very similar to the one shown in the original paper. Note that for data sets of this size, our method uses perplexity and learning rate that are close to the default ones.

**Positioning new points on an existing t-SNE atlas.** A common task in single-cell transcriptomics is to match a given cell to an existing reference data set. For example, introducing a protocol called Patch-seq, ref. [27] performed patch-clamp electro-physiological recordings followed by RNA sequencing of inhibitory cells in layer 1 of mouse visual cortex. Given the existence of the much larger Tasic et al. data set described above, it is natural to ask where on the Fig. 2f, taken as a reference atlas, these Patch-seq cells should be positioned.

It is often claimed that t-SNE does not allow out-of-sample mapping, i.e. no new points can be put on a t-SNE atlas after it is constructed. What this means is that t-SNE is a nonparametric method that does not construct any mapping $f(x)$ from a high-dimensional to the low-dimensional space (parametric t-SNE is possible but is out of scope of this paper, see Discussion). Nevertheless, there is a straightforward way to position a new $x$ on an existing t-SNE atlas. For each Cadwell et al. cell ($n = 46$), we found its $k = 10$ nearest neighbours among the Tasic et al. reference cells, using Pearson correlation across the log-transformed counts of the most variable Tasic et al. genes as distance[28]. Then we positioned the cell at the median t-SNE location of these $k$ reference cells (Fig. 5a). The result agreed very well with the assignment of Cadwell et al. cells to the Tasic et al. clusters performed in ref. [3].

An important caveat is that this method assumes that for each new cell there are cells of the same type in the reference data set. Cells that do not have a good match in the reference data can end up positioned in a misleading way. However, this assumption is justified whenever cells are mapped to a comprehensive reference atlas covering the same tissue, as in the example case shown here.

In a more sophisticated approach[24,29,30], each new cell is initially positioned as outlined above but then its position is optimised using the t-SNE loss function: the cell is attracted to its nearest neighbours in the reference set, with the effective number of nearest neighbours determined by the perplexity. We found that the simpler procedure without this additional optimisation step worked well for our data; the additional optimisation usually has only a minor effect[30].

We can demonstrate the consistency of our method by a procedure similar to a leave-one-out cross-validation. We repeatedly removed one random Tasic et al. cell from the *Vip/Lamp5* clusters, and positioned it back on the same reference t-SNE atlas (excluding the same cell from the $k = 10$ nearest neighbours). Across 100 repetitions, the average distance between the original cell location and the test positioning was $3.2 \pm 2.4$ (mean ± SD; see Fig. 5b for a scale bar), and most test cells stayed within their clusters.

Positioning uncertainty can be estimated using bootstrapping across genes (inspired by ref. [3]). For each of the Patch-seq cells, we repeatedly selected a bootstrapped sample from the set of highly variable genes and repeated the positioning procedure (100 times). This yielded a set of bootstrapped mapping locations; the larger the variability in this set, the larger the uncertainty. To

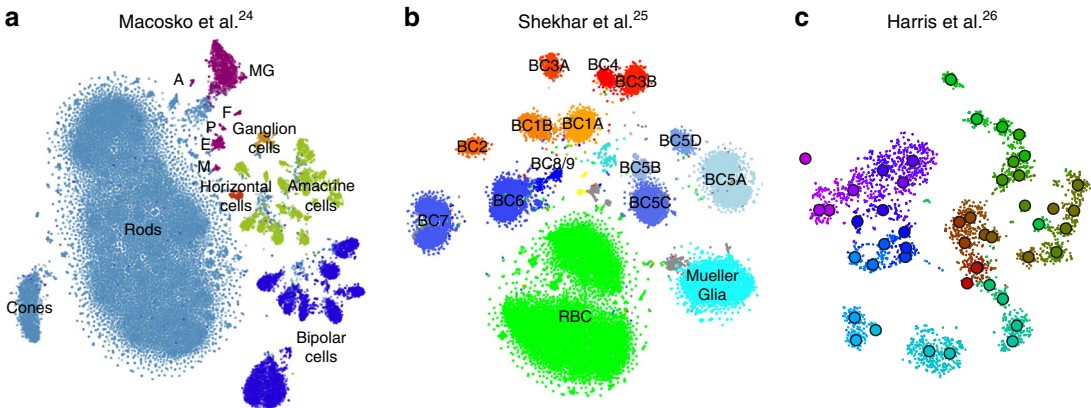

**a** Macosko et al.[24]  **b** Shekhar et al.[25]  **c** Harris et al.[26]

**Fig. 4** UMI-based data sets. Cluster assignments and cluster colours are taken from the original publications. **a** Macosko et al.[24], n = 44,808 cells from the mouse retina. Bipolar cells comprise eight clusters, amacrine cells comprise 21 clusters. Non-neural clusters are abbreviated (MG Mueller glia, A astrocytes, F fibroblasts, P pericytes, E endothelium, M microglia). **b** Shekhar et al.[25], n = 27,499 cells from the mouse retina, mostly bipolar cells. BC bipolar cell, RBC rod bipolar cell. Putative doublets/contaminants shown in grey. Yellow: rod and cone photoreceptors; cyan: amacrine cells. Some clusters appear to consist of two parts; this is due to an experimental batch effect that we did not remove. **c** Harris et al.[26], n = 3663 hippocampal interneurons. Circles denote cluster centroids. The centroid of one cluster (*Sst Cryab*) is not shown because its cells were scattered all over the embedding (as they did in the original publication as well). Cluster labels not shown for visual clarity.

visualise the uncertainty, we show a convex hull covering 95% of the bootstrap repetitions (Fig. 5c), which can be interpreted as a 2D confidence interval. A large polygon means high uncertainty; a small polygon means high precision. For some cells the polygons are so small that they are barely visible in Fig. 5c. For some other cells the polygons are larger and sometimes spread across the border of two adjacent clusters. This suggests that the cluster assignments for these cells are not certain.

**Aligning two t-SNE visualisations.** Tasic et al.[3] is a follow-up to Tasic et al.[31] where n = 1679 cells from mouse visual cortex were sequenced with an earlier sequencing protocol. If one excludes from the new data set all clusters that have cells mostly from outside of the visual cortex, then the remaining data set has n = 19,366 cells. How similar is the cluster structure of this newer and larger data set compared to the older and smaller one? One way to approach this question is through aligned t-SNE visualisations.

To obtain aligned t-SNE visualisations, we first performed t-SNE of the older data set[31] using PCA initialisation and perplexity 50 (Fig. 6a). We then positioned cells of the newer data set[3] on this reference using the procedure described above and used the resulting layout as initialisation for t-SNE (with learning rate $n/12$ and perplexity combination of 30 and $n/100$, as elsewhere). The resulting t-SNE embedding is aligned to the previous one (Fig. 6b).

Several observations are highlighted in Fig. 6. (1) and (2) are examples of well-isolated clusters in the 2016 data that remained well-isolated in the 2018 data (*Sst Chodl* and *Pvalb Vipr2*; here and below we use the 2018 nomenclature). (3) is an example of a small group of cells that was not assigned to a separate cluster back in 2016, became separate on the basis of the 2018 data, but in retrospect appears well-isolated already in the 2016 t-SNE plot (two *L5 LP VISp Trhr* clusters). Finally, (4) shows an example of several clusters in the 2016 data merging together into one cluster based on the 2018 data (*L4 IT VISp Rspo1*). These observations are in good correspondence with the conclusions of ref. [3], but we find that t-SNE adds a valuable perspective and allows for an intuitive comparison.

**Performing t-SNE on large data sets.** Large data sets with n ≫ 100, 000 present several additional challenges to those already

discussed above. First, vanilla t-SNE[9] is slow for n ≫ 1000 and computationally unfeasible for n ≫ 10,000 (see Methods). A widely used approximation called Barnes-Hut t-SNE[32] in turn becomes very slow for n ≫ 100,000. For larger data sets a faster approximation scheme is needed. This challenge was effectively solved by ref. [16] who developed a novel t-SNE approximation called FIt-SNE, based on an interpolation scheme accelerated by the fast Fourier transform. Using FIt-SNE, we were able to process a data set with 1 million points and 50 dimensions (perplexity 30) in 29 min on a computer with four 3.4 GHz double-threaded cores, and in 11 min on a server with twenty 2.2 GHz double-threaded cores.

Second, for n ≫ 100,000, t-SNE with default optimisation parameters tends to produce poorly converged solutions and embeddings with continuous clusters fragmented into several parts. Various groups[16,23] have noticed that these problems can be alleviated by increasing the number of iterations, the length or strength of the early exaggeration (see Methods), or the learning rate. Ref. [15] demonstrated in a thorough investigation that dramatically increasing the learning rate from the default value $\eta = 200$ to $\eta = n/12$ (where 12 is the early exaggeration coefficient[33]) prevents cluster fragmentation during the early exaggeration phase and yields a well-converged solution within the default 1000 iterations.

Third, for n ≫ 100,000, t-SNE embeddings tend to become very crowded, with little white space even between well-separated clusters[18]. The exact mathematical reason for this is not fully understood, but the intuition is that the default perplexity becomes too small compared to the sample size, repulsive forces begin to dominate, and clusters blow up and coalesce like adjacent soap bubbles. While so far there is no principled solution for this in the t-SNE framework, a very practical trick suggested by ref. [34] is to increase the strength of all attractive forces by a small constant exaggeration factor between 1 and ~10 (see Methods). This counteracts the expansion of the clusters.

Fourth, our approach to preserve global geometry relies on using large perplexity $n/100$ and becomes computationally unfeasible for n ≫ 100,000 because FIt-SNE runtime grows linearly with perplexity. For such sample sizes, the only practical possibility is to use perplexity values in the standard range 10–100. To address this challenge, we make an assumption that global geometry should be detectable even after strong

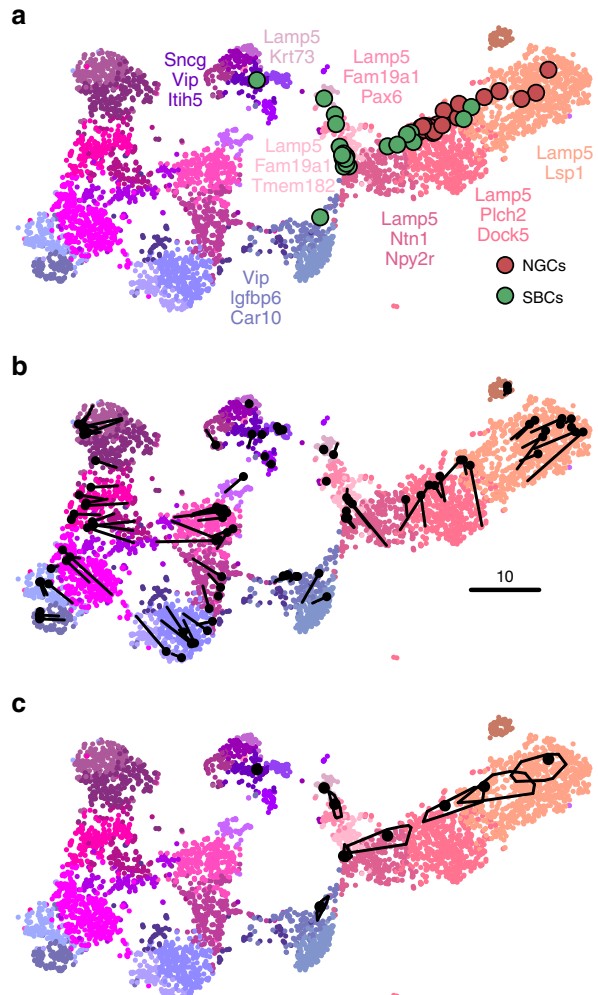

**Fig. 5** Out-of-sample mapping. **a** Interneurons from ref. [27] positioned on the reference t-SNE atlas[3] from Fig. 2f. Only *Vip/Lamp5* continent from Fig. 2f is shown here, as no cells mapped elsewhere. Cluster labels are given only for clusters where at least one cell mapped to. NGC neurogliaform cells; SBC single bouquet cells. Two cells out of 46 are not shown because they had an ambiguous class assignment. **b** A cross-validation approach: 100 random *Vip/Lamp5* cells from the Tasic et al. data set were removed from the t-SNE embedding and then positioned back on it using our method. Black dots show new positions, black lines connect them to the original t-SNE locations of the same cells. **c** Positioning uncertainty for several exemplary cells from panel **a**. Polygons are convex hulls covering 95% of bootstrap repetitions.

downsampling of the data set. This suggests the following pipeline: (i) downsample a large data set to some manageable size; (ii) run t-SNE on the subsample using our approach to preserve global geometry; (iii) position all the remaining points on the resulting t-SNE plot using nearest neighbours; (iv) use the result as initialisation to run t-SNE on the whole data set.

We demonstrate these ideas using two currently largest scRNA-seq data sets. The first one is a 10x Genomics data set with $n = 1{,}306{,}127$ cells from mouse embryonic brain. We first created a t-SNE embedding of a randomly selected subset of $n = 25{,}000$ cells (Fig. 7a). As above, we used PCA initialisation, perplexity combination of 30 and $n/100 = 250$, and learning rate $n/12$. We then positioned all the remaining cells on this t-SNE embedding using their nearest neighbours (here we used Euclidean distance in the PCA space, and $k = 10$ as above; this

took ~10 min). Finally, we used the result as initialisation to run t-SNE on all points using perplexity 30, exaggeration coefficient 4, and learning rate $n/12$ (Fig. 7b).

To validate this procedure, we identified meaningful biological structures in the embedding using developmental marker genes[35–37]. The left part of the main continent is composed of radial glial cells expressing *Aldoc* and *Slc1a3* (Fig. 8a). The neighbouring areas consist of neural progenitors (neuroblasts) expressing *Eomes*, previously known as *Tbr2* (Fig. 8b). The right part of the main continent consists of mature excitatory neurons expressing pan-neuronal markers such as *Stmn2* and *Tubb3* (Fig. 8c) but not expressing inhibitory neuron markers *Gad1* or *Gad2* (Fig. 8d), whereas the upper part of the embedding is occupied by several inhibitory neuron clusters (Fig. 8d). This confirms that our t-SNE embedding shows meaningful topology and is able to capture the developmental trajectories: from radial glia, to excitatory/inhibitory neural progenitors, to excitatory/ inhibitory mature neurons.

We illustrate the importance of the components of our pipeline by a series of control experiments. Omitting exaggeration yielded over-expanded clusters and less discernible global structure (Fig. 7c). Without downsampling, the global geometry was preserved worse (Fig. 7d): e.g. most of the interneuron clusters are in the lower part of the figure, but clusters 17 and 19 (developing interneurons) are located in the upper part. Finally, the default t-SNE with random initialisation and no exaggeration (but learning rate set to $\eta = 1000$) yielded a poor embedding that fragmented some of the clusters and misrepresented global geometry (Fig. 7e). Indeed, overlaying the same marker genes showed that developmental trajectories were not preserved and related groups of cells, e.g. interneurons, were dispersed across the embedding (Fig. 8e–h). Again, this is not a strawman: this embedding is qualitatively similar to the ones shown in the literature[23,38].

In addition, we analysed a data set encompassing $n = 2{,}058{,}652$ cells from mouse embryo at several developmental stages[8]. The original publication showed a t-SNE embedding that we reproduced in Fig. 9a. Whereas it showed a lot of structure, it visibly suffered from all the problems mentioned above: some clusters were fragmented into parts (e.g. clusters 13 and 15), there was little separation between distinct cell types, and global structure was grossly misrepresented. The authors annotated all clusters and split them into ten biologically meaningful developmental trajectories; these trajectories were intermingled in their embedding. In contrast, our t-SNE embedding (Fig. 9b) neatly separated all ten developmental trajectories and arranged clusters within major trajectories in a meaningful developmental order: e.g. there was a continuous progression from radial glia (cluster 7), to neural progenitors (9), to postmitotic premature neurons (10), to mature excitatory (5) and inhibitory (15) neurons.

**Comparison with UMAP**. A promising dimensionality reduction method called UMAP[10] has recently attracted considerable attention in the transcriptomics community[11]. Technically, UMAP is very similar to an earlier method called largeVis[39], but ref. [10] provided a mathematical foundation and a convenient Python implementation. LargeVis and UMAP use the same attractive forces as t-SNE does, but change the nature of the repulsive forces and use a different, sampling-based approach to optimisation. UMAP has been claimed to be faster than t-SNE and to outperform it in terms of preserving the global structure of the data[10,11].

While UMAP is indeed much faster than Barnes-Hut t-SNE, FIt-SNE[16] is at least as fast as UMAP. We found FIt-SNE 1.1 with default settings to be ~4 times faster than UMAP 0.3 with default settings when analysing the 10x Genomics (14 m vs. 56 m) and

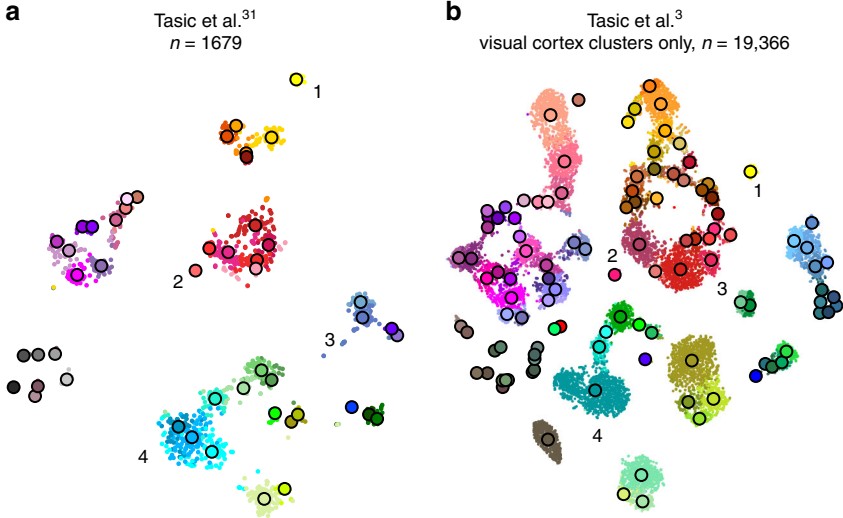

**Fig. 6** Aligned embeddings. **a** T-SNE visualisation of the data set from ref. [31]. Cluster assignments and cluster colours are taken from the original publication. Circles show cluster centroids. Numbers highlight some noteworthy cases, see text. **b** T-SNE visualisation of the data set from ref. [3] after excluding all clusters that mostly consisted of cells from anterior lateral motor cortex (23 clusters that had "ALM" in the cluster name). This t-SNE analysis was initialised by positioning all cells on the reference atlas from panel **a**, ensuring that the two panels are aligned with each other.

the Cao et al. (31 m vs. 126 m) data sets on a server with twenty 2.2 GHz double-threaded cores (for this experiment, the input dimensionality was 50 and the output dimensionality was 2; UMAP may be more competitive in other settings). That said, the exact runtime will depend on the details of implementation, and both methods may be further accelerated in future releases, or by using GPU parallelisation[40].

To compare UMAP with our t-SNE approach in terms of preservation of global structure, we first ran UMAP on the synthetic and on the Tasic et al.[3] data sets (Supplementary Fig. 2). We used the default UMAP parameters, and also modified the two key parameters (number of neighbours and tightness of the embedding) to produce a more t-SNE-like embedding. In both cases and for both data sets, all three metrics (KNN, KNC, and CPD) were considerably lower than with our t-SNE approach. Notably, we observed that in some cases the global structure of UMAP embeddings strongly depended on the random seed. Next, we applied UMAP with default parameters to the 10x Genomics and the Cao et al. data sets. Here UMAP embeddings were qualitatively similar to our t-SNE embeddings, but arguably misrepresented some aspects of the global topology (Supplementary Fig. 3).

An in-depth comparison of t-SNE and UMAP is beyond the scope of our paper, but this analysis suggests that previous claims that UMAP vastly outperforms t-SNE[11] might have been partially due to t-SNE being applied in a suboptimal way. Our analysis also indicates that UMAP does not necessarily solve t-SNE's problems out of the box and might require as many careful parameter and/ or initialisation choices as t-SNE does. Many recommendations for running t-SNE that we made in this manuscript can likely be adapted for UMAP.

## Discussion
The fact that t-SNE does not always preserve global structure is one of its well-known limitations[12]. Indeed, the algorithm, by construction, only cares about preserving local neighbourhoods. We showed that using informative initialisation (such as PCA initialisation, or downsampling-based initialisation) can substantially improve the global structure of the final embedding because it survives through the optimisation process. Importantly, a custom initialisation does not interfere with t-SNE optimisation

and does not yield a worse solution compared to a random initilisation used by default (Fig. 3a, b). A possible concern is that a custom initialisation might bias the resulting embedding by injecting some artefact global structure. However, if anything can be seen as injecting artefactual structure, it is rather the random initialisation: the global arrangement of clusters in a standard t-SNE embedding often strongly depends on the random seed.

We also showed that using large perplexity values ($\sim$1% of the sample size)—substantially larger than the commonly used ones —can be useful in the scRNA-seq context. Our experiments suggest that whereas PCA initialisation helps preserving the macroscropic structure, large perplexity (either on its own or as part of a perplexity combination) helps preserving the mesoscopic structure (Fig. 3d, f).

It has recently been claimed that UMAP preserves the global geometry better than t-SNE[11]. However, UMAP operates on the $k$-nearest neighbour graph, exactly as t-SNE does, and is therefore not designed to preserve large distances any more than t-SNE. To give a specific example, ref. [8], performed both t-SNE and UMAP and observed that "unlike t-SNE, UMAP places related cell types near one another". We demonstrated that this is largely because t-SNE parameters were not set appropriately. Simply using high learning rate $n/12$ places related cell types near one another as well as UMAP does, and additionally using exaggeration factor 4 separates clusters into more compact groups, similar to UMAP.

T-SNE is often perceived as having only one free parameter to tune, perplexity. Under the hood, however, there are also various optimisation parameters (such as the learning rate, the number of iterations, early exaggeration factor, etc.) and we showed above that they can have a dramatic effect on the quality of the visualisation. Here we have argued that exaggeration can be used as another useful parameter when embedding large data sets. In addition, while the low-dimensional similarity kernel in t-SNE has traditionally been fixed as the t-distribution with $\nu = 1$ degree of freedom, we showed in a parallel work that modifying $\nu$ can uncover additional fine structure in the data[41].

One may worry that this gives a researcher too many knobs to turn. However, here we gave clear guidelines on how to set these parameters for effective visualisations. As argued above and in ref. [15], setting the learning rate to $n/12$ ensures good convergence

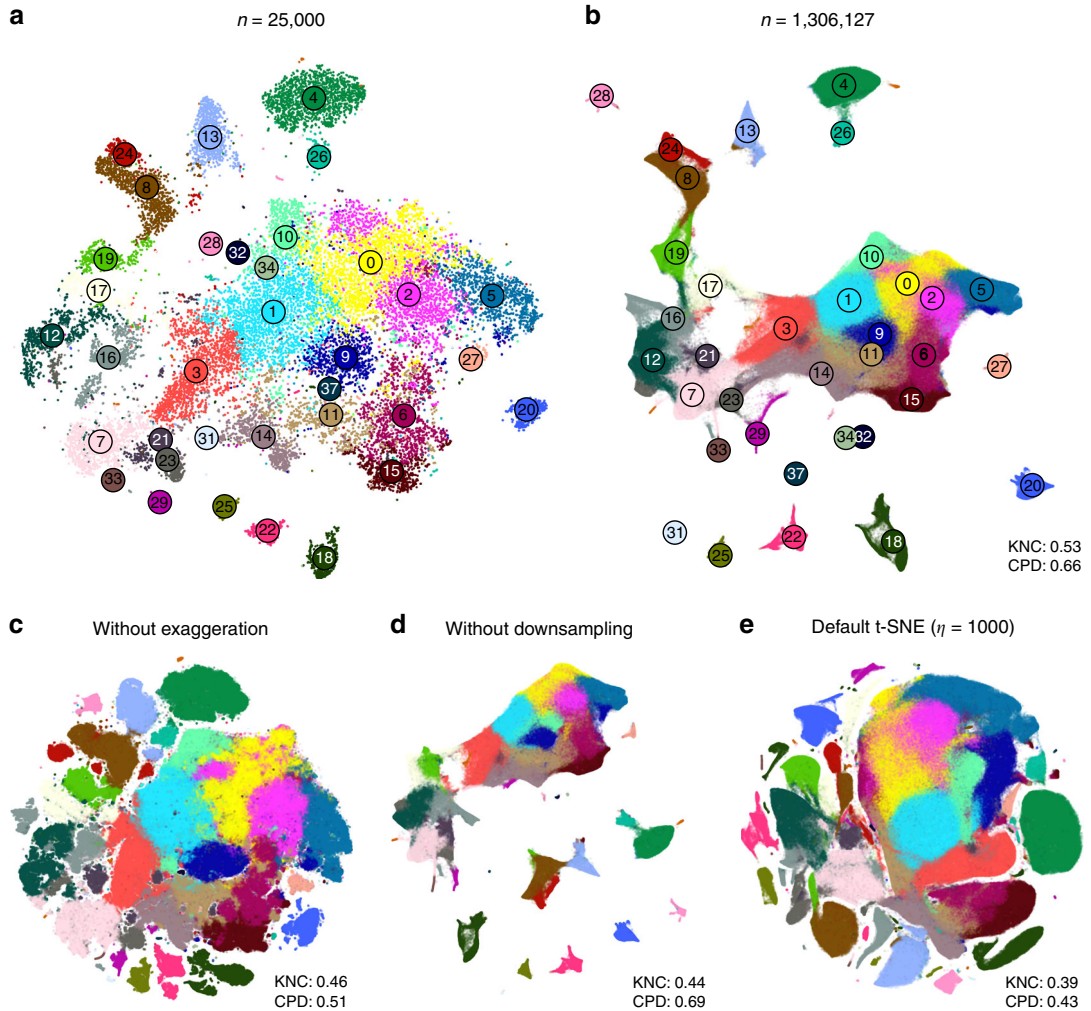

**Fig. 7** 10x Genomics data set. Sample size $n = 1,306,127$. Cluster assignments and cluster colours are taken from ref. [23]. **a** T-SNE of a random subsample of 25,000 cells (PCA initialisation, perplexity combination of 30 and 250, learning rate 25,000/12). Cluster labels for several small clusters (30, 35, 36, and 38) are not shown here and in **b** because these clusters were very dispersed in the embeddings. **b** T-SNE of the full data set. All cells were positioned on the embedding in panel **a** and this was used as initialisation. Perplexity 30, exaggeration 4, learning rate $n/12$. **c** The same as in **b** but without exaggeration. **d** The same as in **b** but with PCA initialisation, i.e. without using the downsampling step. **e** Default t-SNE with learning rate set to $\eta = 1000$: random initialisation, no exaggeration.

and automatically takes care of the optimisation issues. Perplexity should be left at the default value 30 for very large data sets, but can be combined with $n/100$ for smaller data sets. Exaggeration can be increased to ∼4 for very large data sets, but is not needed for smaller data sets.

In comparison, UMAP has two main adjustable parameters (and many further optimisation parameters): n_neighbors, corresponding to perplexity, and min_dist, controlling how tight the clusters become. The latter parameter sets the shape of the low-dimensional similarity kernel[10] and is therefore analogous to $\nu$ mentioned above. Our experiments with UMAP suggest that its repulsive forces roughly correspond to t-SNE with exaggeration ∼4 (Supplementary Figs. 2, 3). Whether this is desirable, depends on the application. With t-SNE, one can choose to switch exaggeration off and e.g. use the embedding shown in Fig. 7c instead of Fig. 7b.

Several variants of t-SNE have been recently proposed in the literature. One important example is parametric t-SNE, where a neural network is used to create a mapping $f(\boldsymbol{x})$ from high-dimensional input space to two dimensions and is trained using standard deep learning techniques to yield an optimal t-SNE result[42]. Parametric t-SNE has been recently applied to

transcriptomic data under the names net-SNE[43] and scvis[44]. The latter method combined parametric t-SNE with a variational autoencoder, and was claimed to yield more interpretable visualisations than standard t-SNE due to better preserving the global structure. Indeed, the network architecture limits the form that the mapping $f(\boldsymbol{x})$ can take; this implicit constraint on the complexity of the mapping prevents similar clusters from ending up in very different parts of the resulting embedding. Also, in this approach the most appropriate perplexity does not need to grow with the sample size, as long as the mini-batch size remains constant. By default scvis uses mini-batch size of 512 and perplexity 10, which likely corresponds to the effective perplexity of $10/512 \cdot n$, i.e. ∼2% of the sample size, similar to our 1% suggestion here.

Another important development is hierarchical t-SNE, or HSNE[45]. The key idea is to use random walks on the $k$-nearest neighbours graph of the data to find a smaller set of landmarks, which are points that can serve as representatives of the surrounding points. In the next round, the $k$-nearest neighbours graph on the level of landmarks is constructed. This operation can reduce the size of the data set by an order of magnitude, and can be repeated until the data set becomes small enough to be

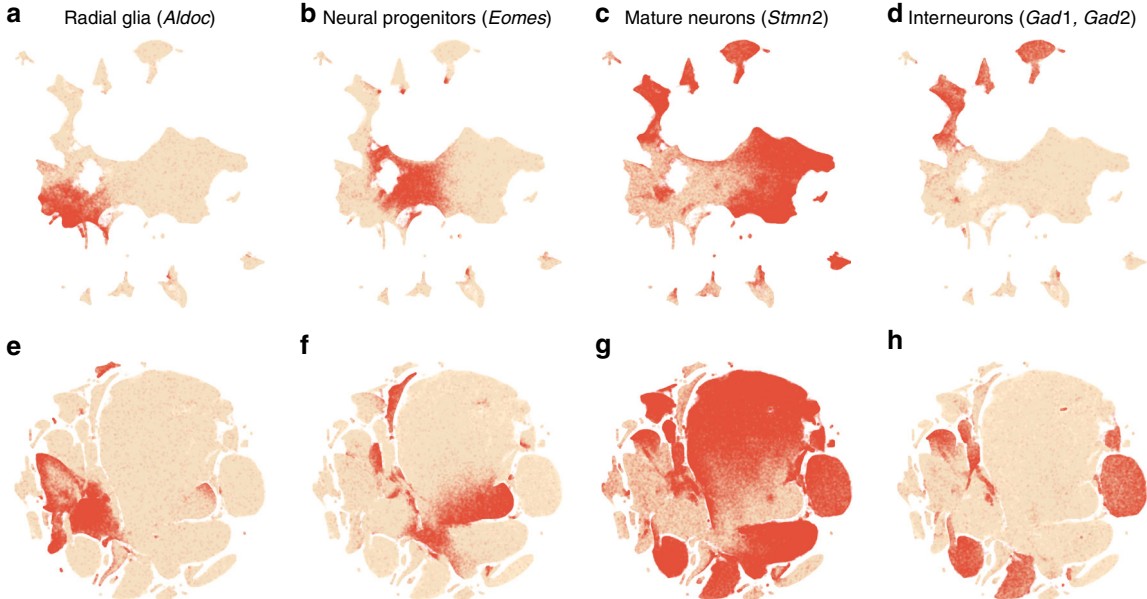

**Fig. 8** Developmental marker genes. Overlay over t-SNE embeddings from Fig. 7. **a** Expression of *Aldoc* gene (marker of radial glia) on the t-SNE embedding from Fig. 7b. Any cell with *Aldoc* detected (UMI count above zero) was coloured in red. Another radial glia marker, *Slc1a3*, had similar but a bit broader expression. **b** Expression of *Eomes*, marker of neural progenitors (neuroblasts). **c** Expression of *Stmn2*, marker of mature neurons. A pan-neuronal marker *Tubb3* had similar but a bit broader expression. **d** Expression of *Gad1* and *Gad2* (either of them), markers of inhibitory neurons. **e**–**h** The same genes overlayed over the default t-SNE embedding from Fig. 7e.

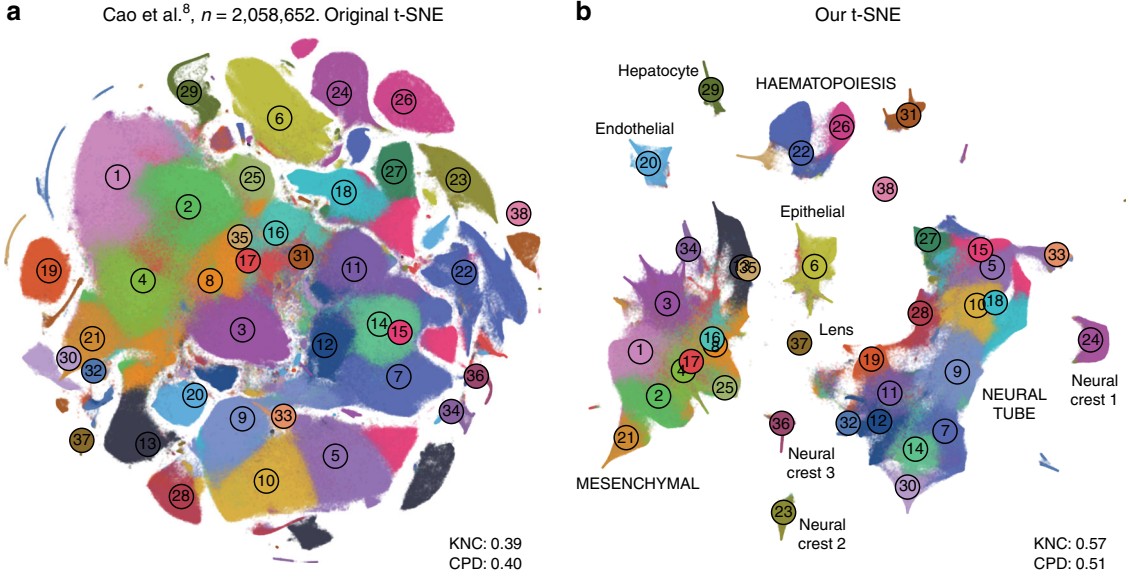

**Fig. 9** Cao et al. data set. Sample size $n = 2,058,652$. Cluster assignments and cluster colours are taken from the original publication[8]. **a** T-SNE embedding from the original publication. The authors ran t-SNE in `scanpy` with default settings, i.e. with random initialisation, perplexity 30, and learning rate 1000. For cluster annotations, see original publication. **b** T-SNE embedding produced with our pipeline for large data sets: a random sample of 25,000 cells was embedded using PCA initialisation, learning rate 25,000/12, and perplexity combination of 30 and 250; all other cells were positioned on resulting embedding and this was used to initialise t-SNE with learning rate 2,058,652/12, perplexity 30, and exaggeration 4. Labels correspond to the ten developmental trajectories identified in the original publication. Labels in capital letters denote trajectories consisting of multiple clusters. 32,011 putative doublet cells are not shown in either panel.

analysed with t-SNE. Each level of the landmarks hierarchy can be explored separately. Ref. [18] successfully applied this method to mass cytometry data sets with up to 15 million cells. However, HSNE does not allow to embed all *n* points in a way that would preserve the geometry on the level of landmarks.

Our approach to preserving global geometry of the data is based on using PCA initialisation and large perplexity values. It can fail if some aspects of the global geometry are not adequately captured in the first two PCs or by the similarities computed using a large perplexity. This may happen when the data set contains very isolated but rare cell types. Indeed, a small isolated cluster might not appear isolated in the first two PCs because it would not have enough cells to contribute much to the explained variance. At the same time, large perplexity will make the points

in that cluster be attracted to some arbitrary, unrelated, clusters. As a result, a small cluster can get sucked into the middle of the embedding even if it is initialised on the periphery.

This is what happened in the Shekhar et al. data set (Fig. 4b): cone and rod photoreceptor (yellow) and amacrine cell (cyan) clusters ended up in the middle of the embedding despite being very different from all the bipolar cell clusters. This can be seen in the MDS embedding of the cluster means which is unaffected by the relative abundances of the clusters; thus, when t-SNE is done together with clustering, we recommend to supplement a t-SNE visualisation with a MDS visualisation of cluster means (as in Fig. 2a). Alternatively, one could use PAGA[46], a recent method specifically designed to visualise the relationships between clusters in scRNA-seq data.

This example highlights that our approach is not a final solution to preserving the global structure of the data. A principled approach would incorporate some terms ensuring adequate global geometry directly into the loss function, while making sure that the resulting algorithm is scalable to millions of points. We consider it an important direction for future work. In the meantime, we believe that our recommendations will strongly improve t-SNE visualisations used in the current single-cell transcriptomic studies, and may be useful in other application domains as well.

## Methods

**The t-SNE loss function.** The t-SNE algorithm[9] is based on the SNE framework[47]. SNE introduced a notion of directional similarity of point $j$ to point $i$,

$$p_{j|i} = \frac{\exp(-\|\mathbf{x}_i - \mathbf{x}_j\|^2/2\sigma_i^2)}{\sum_{k \neq i} \exp(-\|\mathbf{x}_i - \mathbf{x}_k\|^2/2\sigma_i^2)}, \quad (1)$$

defining, for every given point $i$, a probability distribution over all points $j \neq i$ (all $p_{i|i}$ are set to zero). The variance of the Gaussian kernel $\sigma_i^2$ is chosen such that the perplexity of this probability distribution

$$\mathcal{P}_i = \exp\left(-\log(2) \cdot \sum_{j \neq i} p_{j|i} \log_2 p_{j|i}\right) \quad (2)$$

has some pre-specified value. The larger the perplexity, the larger the variance of the kernel, with the largest possible perplexity value equal to $n - 1$ corresponding to $\sigma_i^2 = \infty$ and the uniform probability distribution ($n$ is the number of points in the data set). Importantly, for any given perplexity value $\mathcal{P}$, all but $\sim \mathcal{P}$ nearest neighbours of point $i$ will have $p_{j|i}$ very close to zero. For mathematical and computational convenience, symmetric SNE defined undirectional similarities

$$p_{ij} = \frac{p_{i|j} + p_{j|i}}{2n}, \quad (3)$$

such that $\sum_{i,j} p_{ij} = 1$, i.e. this is a valid probability distribution on the set of all pairs $(i, j)$.

The main idea of SNE and its modifications is to arrange the $n$ points in a low-dimensional space such that the similarities $q_{ij}$ between low-dimensional points match $p_{ij}$ as close as possible in terms of the Kullback-Leibler divergence. The loss function is thus

$$\mathcal{L} = \sum_{i,j} p_{ij} \log \frac{p_{ij}}{q_{ij}}. \quad (4)$$

The main idea of t-SNE was to use a t-distribution with one degree of freedom (also known as Cauchy distribution) as the low-dimensional similarity kernel:

$$q_{ij} = \frac{w_{ij}}{Z}, \quad w_{ij} = \frac{1}{1 + \|\mathbf{y}_i - \mathbf{y}_j\|^2}, \quad Z = \sum_{k \neq l} w_{kl}, \quad (5)$$

where $\mathbf{y}_i$ are low-dimensional coordinates (and $q_{ii} = 0$). As a matter of definition, we consider any method that uses the t-distribution as the output kernel and Kullback-Leibler divergence as the loss function to be t-SNE; similarities $p_{j|i}$ can in principle be computed using non-Euclidean distances instead of $\|\mathbf{x}_i - \mathbf{x}_j\|$ or can use non-perplexity-based calibrations.

To justify our intuitive explanation in terms of attractive and repulsive forces, we can rewrite the loss function as follows:

$$\mathcal{L} = \sum_{i,j} p_{ij} \log \frac{p_{ij}}{q_{ij}} = \text{const} - \sum_{i,j} p_{ij} \log q_{ij}, \quad (6)$$

and dropping the constant,

$$-\sum_{i,j} p_{ij} \log \frac{w_{ij}}{Z} = -\sum_{i,j} p_{ij} \log w_{ij} + \sum_{i,j} p_{ij} \log Z = -\sum_{i,j} p_{ij} \log w_{ij} + \log \sum_{i,j} w_{ij}. \quad (7)$$

To minimise $\mathcal{L}$, the first sum should be as large possible, which means large $w_{ij}$, i.e. small $\|\mathbf{y}_i - \mathbf{y}_j\|$, meaning an attractive force between points $i$ and $j$ whenever $p_{ij} \neq 0$. At the same time, the second term should be as small as possible, meaning small $w_{ij}$ and a repulsive force between any two points $i$ and $j$, independent of the value of $p_{ij}$.

**The t-SNE optimisation.** The original publication[9] suggested optimising $\mathcal{L}$ using adaptive gradient descent with momentum. The authors initialised $\mathbf{y}_i$ randomly, using a Gaussian distribution with standard deviation 0.0001. It is important that initial values have small magnitude: otherwise optimisation often fails to converge to a good solution.

To escape bad local minima and allow similar points to be quickly pulled together, the original publication[9] suggested an "early exaggeration" trick: during initial iterations they multiplied all attractive forces by $\alpha > 1$. Later work[32] used $\alpha = 12$ for the first 250 iterations, which became the default since then.

The exact t-SNE computes $n^2$ similarities $p_{ij}$ and $n^2$ pairwise attractive and repulsive forces on each gradient descent step. This becomes unfeasible for $n \gg 10,000$. A follow-up paper[32] suggested two approximations in order to speed up the computations. First, it noticed that for any perplexity value $\mathcal{P}$ all but $\mathcal{O}(\mathcal{P})$ nearest neighbours of any given point $i$ will have nearly zero values $p_{j|i}$. It suggested to only find $k = 3\mathcal{P}$ nearest neighbours of each point and set $p_{j|i} = 0$ for the remaining $n - 3\mathcal{P}$ points. This relied on finding the exact $3\mathcal{P}$ nearest neighbours, but in later work various authors[10,16,39,48] started using approximate nearest neighbour algorithms which is much faster and does not seem to make t-SNE results any worse.

Using $3\mathcal{P}$ nearest neighbours accelerates computation of the attractive forces. To accelerate the repulsive force computations, ref. [32] used the Barnes-Hut approximation, originally developed for N-body simulations in physics. This reduces computational complexity from $\mathcal{O}(n^2)$ to $\mathcal{O}(n \log n)$, works reasonably fast for $n$ up to $\sim 100,000$, but becomes too slow for much larger sample sizes. Inspired by the fast multipole method, another technique originally developed for N-body simulations, ref. [16] suggested to interpolate repulsive forces on an equispaced grid and to use fast Fourier transform to accelerate the interpolation (FIt-SNE). This lowers computational complexity to $\mathcal{O}(n)$ and works very fast for $n$ into millions. An important limitation is that it is only implemented for 1D and 2D, but not for 3D embeddings, as the interpolation would work much slower in 3D.

**Synthetic data set.** All points were generated from a standard Gaussian distribution in 50 dimensions. All points from each class were shifted by 20 in mutually orthogonal directions. All points from each type in one class ($n = 2000$ per type) were additionally shifted by 4 in mutually orthogonal directions; in another class ($n = 1000$ per type)—by 10; in the third class ($n = 100$ per type)—also by 10. The resulting total sample size was $n = 15,500$.

**Data preprocessing.** All data sets were downloaded as tables of UMI or read counts. Let $\mathbf{X}$ be a $n \times p$ matrix of gene counts, with $n$ and $p$ being the number of cells and the number of genes respectively. We assume that zero columns (if any) have been removed. We used a standard preprocessing pipeline consisting of the following steps: (i) sequencing depth normalisation; (ii) feature selection; (iii) log-transformation; (iv) PCA. Specifically, we normalised the read counts to counts per million (CPM) and UMI counts to counts per median sequencing depth, selected 1000–3000 most variable genes using dropout-based feature selection similar to the one suggested in ref. [49], applied $\log_2(x + 1)$ transform, and finally did PCA retaining the 50 leading PCs. We experimented with modifying and omitting any of these steps. Our experiments showed that log-transformation (or a similar non-linear transformation) and feature selection are the two most important steps for adequate results. PCA mainly improves computational efficiency as it reduces the dimensionality and size of the data set before running t-SNE.

**Sequencing depth normalisation.** We divided the read counts of each cell by the cell's sequencing depth $\sum_k X_{ik}$, and multiplied by 1 million, to obtain CPM. We normalised the UMI counts by the cell's sequencing depth and multiplied by the median sequencing depth across all $n$ cells in the data set[50]. This is more appropriate for UMI counts because multiplying by 1 million can strongly distort the data after the subsequent $\log(1 + x)$ transform[51].

**Feature selection.** Most studies use the mean-variance relationship to perform feature selection: they select genes that have large variance given their mean. We adopt the approach of ref. [49] that exploited the mean-dropout relationship: the idea is to select genes that have high dropout (i.e. zero count) frequency given their mean expression level. Any gene that has high dropout rate and high mean expression could potentially be a marker of some particular subpopulation. We

found it more intuitive to use the mean across non-zero counts instead of the overall mean, because it is computed independently of the fraction of zero counts.

For each gene $g$, we computed the fraction of near-zero counts

$$d_g = \frac{1}{n}\sum_i I(X_{ig} \leq t) \qquad (8)$$

and the mean log non-zero expression

$$m_g = \left\langle \log_2 X_{ig} | X_{ig} > t \right\rangle. \qquad (9)$$

For all UMI-based data sets we used $t = 0$ and for all Smart-seq2/SMARTer data sets we used $t = 32$ (some known marker genes were not getting selected with $t = 0$). We discarded all genes that have non-zero expression in less than $n_{min} = 10$ cells. There was a strong negative relationship between $d_g$ and $m_g$ across all the remaining genes (Supplementary Fig. 4). In order to select a pre-specified number $M$ of genes (usually $M = 1000$ or $M = 3000$), we used a heuristic approach of finding a value $b$ such that

$$d_g > \exp\left[-a(m_g - b)\right] + 0.02 \qquad (10)$$

was true for exactly $M$ genes. This can be done with a binary search. In Supplementary Fig. 4 this corresponds to moving the red line horizontally until there are exactly $M$ genes to the upper-right of it. These $M$ genes were then selected. We used $a = 1.5$ for all data sets apart from Macosko et al. where $a = 1$ provided a better fit for the distribution.

We performed feature selection using the raw counts (before sequencing depth normalisation). Then we used normalised values for the selected $M$ genes. We used $M = 3000$ for the Tasic et al. 2018 and for the Macosko et al. data sets, and $M = 1000$ for the remaining data sets. This feature selection method was not used for the 10x Genomics and the Cao et al. data sets, see below.

**Nonlinear transformation**. We transformed all values in the $n \times M$ count matrix after feature selection with a $\log_2(x + 1)$ transformation. This transformation is standard in the transcriptomics literature. It is convenient because all zeros remain zeros, and at the same time the expression counts of all genes become roughly comparable. Without this transformation, the Euclidean distances between cells are dominated by a handful of genes with very high counts. There are other transformations that can perform similarly well. In the cytometric literature, the inverse hyperbolic sine $\mathrm{arsinh}(x) = \log(x + \sqrt{x^2 + 1})$ is often used, sometimes as $\mathrm{arsinh}(x/r)$ with $r = 5$ or a similar value[17]. Note that $\mathrm{arsinh}(x/r)$ is the variance-stabilising transformation for the negative binomial distribution with parameter $r$, which is often taken to model UMI counts well.

**Standardisation**. Many studies standardise the $n \times M$ matrix after the log-transformation, i.e. centre and scale all columns to have zero mean and unit variance. We prefer not to do this by default. In general, standardisation is recommended when different features are on different scale, but the log-transformed counts of different genes are arguably already on the same scale.

From a more theoretical point of view, if one could assume that the expression counts of each gene for cells of the same type are distributed log-normally, then Euclidean distance after log-transformation would exactly correspond to the log-likelihood. For the UMI-based data, the common assumption is that the expression counts are distributed according to the negative binomial distribution. For large counts, the negative binomial distribution behaves qualitatively similarly to the log-normal (for example, its variance function is $\mu + \mu^2/r$ whereas the log-normal has variance function proportional to $\mu^2$), so the Euclidean distance after the log-transformation can be thought of as approximating the negative binomial log-likelihood[26]. Standardising all the genes after log-transformation would destroy this relationship.

At the same time, in some data sets we observed a stronger separation between some of the clusters after performing the standardisation step. Here we applied standardisation for those data sets in which it was used by the original authors (Macosko et al., Shekhar et al., 10x Genomics, Cao et al.).

`Scanpy` **preprocessing**. For the 10x Genomics and the Cao et al. data sets, we used the preprocessing pipeline `recipe_zheng17()` from `scanpy`[23] to ease the comparison with clustering and dimensionality reduction performed by ref. [23] and ref. [8]. This pipeline follows ref. [50] and is similar to ours: it performs sequencing depth normalisation to median sequencing depth, selects most variable genes based on the mean-variance relationship, applies the $\log_2(x + 1)$ transform and standardises each feature. We used $M = 1000$ genes for the 10x Genomics data set and $M = 2000$ for the Cao et al. data set, following the original publication.

**Principal component analysis**. We used PCA to reduce the size of the data matrix from $n \times M$ to $n \times 50$ prior to running t-SNE. In our experiments, this did not have much influence on the t-SNE results but is computationally convenient. Some studies (e.g. ref. [25]) estimate the number of significant principal components via shuffling. In our experiments, for the data sets with tens of thousands of cells, the number of significant PCs was usually close to 50 (for example, for the Tasic et al.[3]

data set it was 40, according to the shuffling test). Given that PCA does not have much influence on the t-SNE results, we prefer to use a fixed value of 50.

**Default parameters for t-SNE optimisation**. Unless explicitly stated, we used the default parameters of FIt-SNE. Following ref. [32], the defaults are 1000 iterations with learning rate $\eta = 200$; momentum 0.5 for the first 250 iterations and 0.8 afterwards; early exaggeration $\alpha = 12$ for the first 250 iterations; initialisation drawn from a Gaussian distribution with standard deviation 0.0001. Further input parameters for the nearest neighbour search using the `Annoy` library (number of trees: 50, number of query nodes: $3\mathcal{P} \cdot 50$ for perplexity $\mathcal{P}$) and for the grid interpolation (number of approximating polynomials: 3, maximum grid spacing: 1, minimum grid size: 50) were always left at the default values. For reproducibility, we always used random seed 42.

**Initialisation**. For PCA initialisation, we divide the first two principal components by the standard deviation of PC1 and multiply them by 0.0001, which is the default standard deviation of the random initialisation. This scaling is important: values used for initialisation should be close to zero, otherwise the algorithm might have problems with convergence (we learned about the importance of scaling from James Melville's notes at https://jlmelville.github.io/smallvis/init.html). The same scaling was used for the downsampling-based initialisation and also for the custom initialisation when creating aligned visualisations.

The sign of the principal components is arbitrary. To increase reproducibility of the figures, we always fixed the sign of the first two PCs such that for each PCA eigenvector the sum of its values were positive.

**Post-processing**. For the t-SNE embeddings of the 10x Genomics data set, we rotated the result 90 degrees clockwise and flipped horizontally, to make it visually more pleasing. Note that t-SNE result can be arbitrarily rotated and flipped as this does not change the distances between points. Caveat: it should not be stretched horizontally or vertically.

**Multi-scale similarities**. We follow ref. [13] in the definition of multi-scale similarities. For example, to combine perplexities 30 and 300, the values $p_{j|i}$ are computed with perplexity 30 and with perplexity 300 for each cell $i$ and then averaged. This is approximately (but not exactly) equivalent to using a different similarity kernel: instead of the Gaussian kernel $\exp(-d^2/2\sigma_i^2)$ where $d$ is Euclidean distance, a multi-scale kernel

$$\frac{1}{\sigma_i}\exp\left(-\frac{d^2}{2\sigma_i^2}\right) + \frac{1}{\tau_i}\exp\left(-\frac{d^2}{2\tau_i^2}\right), \qquad (11)$$

with the variances $\sigma_i^2$ and $\tau_i^2$ selected such that the perplexity of the first Gaussian component is 30 and the perplexity of the second Gaussian component is 300.

**Learning rate**. Following ref. [15], we used learning rate $\eta = n/12$ as it is defined in FIt-SNE. FIt-SNE (as well as openTSNE) followed the convention of the original[9] and the Barnes-Hut[32] t-SNE implementations and omitted factor 4 in the gradient equation. Some other existing t-SNE implementations such as the one in `scikit-learn` do include the factor 4. There one would need to use $\eta = n/48$ to achieve the same result.

**Exaggeration**. Early exaggeration means multiplying the attractive term in the loss function (Eq. (7)) by a constant $\alpha > 1$ during the initial iterations (the default is $\alpha = 12$ for 250 iterations). Ref. [34] suggested to use late exaggeration: using some $\alpha > 1$ for some number of last iterations. Thus, their approach used three stages: early exaggeration, followed by no exaggeration, followed by late exaggeration. We prefer to use two stages only: we keep $\alpha$ constant after the early exaggeration is turned off. This is why we simply call it exaggeration and not late exaggeration. While early exaggeration is essentially an optimisation trick[9], we consider subsequent exaggeration to be a meaningful change of the loss function.

**Positioning out-of-sample cells**. To position new cells on an existing t-SNE embedding we used the same $M$ most variable genes that were used to create the target embedding. Usually only a subset of $L < M$ genes was present in the count table of the new data set; we used $\log_2(1 + x)$-transformed counts of these $L$ genes to compute the correlation distances. To position a cell, we used coordinate-wise median among its $k$ nearest neighbours.

We used correlation distances as possibly more robust for batch effects than Euclidean distances: when out-of-sample cells are sequenced with a different protocol, the batch effect can be very strong. This consideration does not apply to positioning cells for a downsampling-based initialisation (without any possible batch effect). For computational simplicity, here we used Euclidean distance in the space of the 50 PCs.

**Bootstrapping over genes**. We used bootstrapping to estimate the uncertainty of the mapping of new cells to the existing t-SNE atlas. Given a set of $L$ genes that are used for the mapping, we selected a bootstrap sample of $L$ genes out of $L$ with

repetition and performed the positioning procedure using this sample of genes. This constitutes one bootstrap iteration. We did 100 iterations and, for each cell, obtained 100 positions on the t-SNE atlas. The larger the spread of these positions, the larger the mapping uncertainty.

For Fig. 5c, we computed the distances from the original mapping position to the 100 bootstrapped positions and discarded five bootstrap positions with the largest distance. Then we drew a convex hull of the remaining 95 bootstrap positions (using `scipy.spatial.ConvexHull`).

**Reporting summary**. Further information on research design is available in the Nature Research Reporting Summary linked to this article.

## Data availability

All data were downloaded in the form of count tables following links in the original publications.

## Code availability

We prepared a self-contained Jupyter notebook in Python that demonstrates all the techniques presented in this manuscript. It is available at https://github.com/berenslab/rna-seq-tsne. For simplicity, it uses the Tasic et al.[3] data set for all demonstrations. To show how to map new cells to the reference t-SNE atlas, we split the data into a training set and a test set. To show how to align two t-SNE visualisations, we split the data set into two parts. To show how to process a much larger data set, we replicate each cell 10 times and add noise.

The code that does the analysis and produces all the figures used in this manuscript is available in form of Python notebooks at https://github.com/berenslab/rna-seq-tsne. We used a C++ implementation of FIt-SNE[16], version 1.1, available at https://github.com/klugerlab/FIt-SNE together with interfaces for R, Matlab, and Python. While working on this paper, we contributed to this package several additional features that were crucial for our pipeline.

FIt-SNE has been re-implemented as a pure Python package openTSNE[52], available at https://github.com/pavlin-policar/openTSNE. It supports all the features used in this manuscript (and conveniently allows us to position out-of-sample cells, with or without optimisation).

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

## Acknowledgements

The authors thank Anna Belkina, George Linderman, Leland McInnes, James Melville, Pavlin Poličar, and Alexander Wolf for discussions of t-SNE and UMAP. The authors thank Andreas Tolias for discussing Patch-seq data. This work was supported by the Deutsche Forschungsgemeinschaft (BE5601/4-1; Cluster of Excellence "Machine Learning—New Perspectives for Science", EXC 2064, project number 390727645), the Federal Ministry of Education and Research (FKZ 01GQ1601 and 01IS18039A) and the National Institute of Mental Health of the National Institutes of Health under Award Number U19MH114830. The content is solely the responsibility of the authors and does not necessarily represent the official views of the National Institutes of Health.

## Author contributions

D.K. and P.B. conceptualised the project, D.K. performed the analysis, D.K. and P.B. wrote the paper.

## Competing interests

The authors declare no competing interests.
