## [Peer Review File · Nature Communications]

Reviewers' comments:

Reviewer #1 (Remarks to the Author):

The manuscript titled "The art of using t-SNE for single-cell transcriptomics" presents a protocol for using t-SNE for single-cell transcriptomics. It is well written and easy to follow. The authors propose a combination of preprocessing and initial dimensionality reduction of the data with PCA followed different custom initializations (by means of PCA or a t-SNE with fewer points, which can also be used to align datasets) with multiple perplexity values for the actual t-SNE computation to conserve global distances.

While the results in many cases look very promising and the protocol might be a useful addition to the tool chain for analyzing single-cell transcriptomics, there is little to no formal evaluation and it seems quite ad-hoc. Arguably these techniques are hard to evaluate since we do have little understanding of the contents of given complex, high-dimensional data. However, we must be rigorous in the evaluation of our tools. I would propose to at least add a synthetic dataset that is well understood and do proper parameter space testing and quantitative evaluation of the results for the shown data.

Main concerns:

I think the title does not fit the content of the paper. Besides strongly disagreeing with the phrase art (see below), after reading the title I expected and hoped for an explanation on how to use and interpret t-SNE embeddings that seems to be urgently needed by the community.

The main text then, however, describes yet another protocol incorporating some preprocessing and (seemingly) ad-hoc parameterization.

While superficially, the visualizations presented in the results mostly look nice, there is no quantitative evaluation and even the qualitative evaluation is very superficial. Several times, the authors make strong statements that one or the other embedding is better with very little and vague to no justification.

E.g. p6 "In fact, perplexity 500 yields a surprisingly good visualisation", "The result, when initialised with PCA, is arguably the best t-SNE visualisation of this data set that we were able to obtain."

p7 "Our t-SNE visualisation (Figure 4c) did not use any of that but was nevertheless at least as good as the original one."

While the authors to sometimes point out some reasons why they deem one or the other result 'better' it is very hand-wavy and unstructured.

In some cases I also do not agree with the assessment. For example let's consider Figure 5. To me, 5e seems like a rather good result (considering that plotting 1M points in a small rectangle is bound to result in dense maps). I see a lot of structure, why is issue #2 worse here than in the others, there is clear separation between clusters?

Furthermore, overlaying the cluster colors gives a false impression when judging the quality. This information is not there when exploring a new dataset so we could only judge by structure or overlaying genes as in Figure 8.

In general there should be a much more structured analysis of these results.

What are the metrics to judge the quality of an embedding? How do we know the structure claimed to be better is not imposed by some of the preprocessing? Especially the initialization has great impact on the final embedding. But how do we know that it does not enforce structure that is not there?

The last point brings me to another concern. There is no formal evaluation of which structure is actually in the data. Then how can we know that structure in the t-SNE embedding is actually coming from the data and which is coming from the initialization. Especially initializing with a random sample has a danger of placing 'unseen' types in a structure they don't belong. And these unseen types are supposedly of major interest.

* How is this tackled especially when downsampling to create the initial t-SNE?

The authors show an example (of 10 cells?) in Figure 5, some of these seemingly show very large uncertainty/error, even excluding the biggest outliers. Also basically all of these examples are in the same major area of the embedding what happened in the other areas? Again, a quantitative evaluation is necessary.

* What downsampling approach is used? Random? What effect does this have on rare types that might get lost during downsampling?

* When used for point placement only, again, what happens with 'unseen' types when further optimization, as proposed by Berman et al (2014), is omitted?

Finally, in basically all of the steps of the protocol several parameters are needed, from the number of PCA dimensions to a plethora of parameters for the t-SNE itself. There is little justification for any of the used values except for that they are the standard values, or that the result was good and no analysis how changing the parameters impacts the results. I would like to refer the authors to Belkina et al. <https://doi.org/10.1101/451690> as an example of rigorous parameter analysis.

Furthermore, for all of the presented datasets these parameters/and sometimes initialization method are different. This is not helpful. For the protocol to be useful in practice in the best case we would have robust parameters that do not need much tuning, but at least we need guidelines on how to adjust these parameters to a given dataset.

Other:

In my opinion the term art and the authors claim that "In data visualization, as a matter of principle, there is just no "one size fits all" solution — the reason we call this "the art of using t-SNE"." are strongly misleading.

While indeed there are rarely one size fits all solutions in visualization (or other computational sciences for that matter) a proper visualization (-method) is a result of rigorous analysis and a solid theoretical foundation and justification. As laid out above, most of this is missing in the paper.

Some computational performance numbers would be helpful. Some of the proposed techniques seem quite expensive (large perplexity, nearest neighbor search for point placement (I assume hence the $k=1$ for large datasets), etc.)

p4. "In general, inspecting initial t-SNE results often allows to discover small outlying clusters of cell doublets or other experimental artefacts. We recommend discarding such cells and re-running the analysis: otherwise such artefact clusters can create "bridges" between unrelated t-SNE clusters and spoil the visualisation."

How do we know those bridges are artifacts and not signal. And if they could be signal aren't those bridges actually helpful to keep global structure?

p5. "we will, from now on, not use cluster labels for any analysis."

This is a bit confusing. I assume the authors mean they do not use the cluster labels as input for the t-SNE (they use them for the visuals in basically all figures). Does this mean for Figure 2 they used the labels as input?

p6. "perplexity 500 yields a surprisingly good visualisation."

Is this really surprising? Isn't that exactly what we know a larger perplexity does?

p11. "Validating this procedure is not straightforward, as there are no annotations available for the clusters."

Weren't these used in Figure 7 for coloring?

Reviewer #2 (Remarks to the Author):

The authors investigated the various parameters of the t-distributed Stochastic Neighbor Embedding (t-SNE) algorithm in the context of several scRNAseq data sets, the largest of which included 1.3 million points. They integrate two existing heuristics, PCA initialization and multi-scale combination of perplexity values, and show that these lead to improved t-SNE results. Additionally, the authors show that k-nearest neighbors classification can be used to embed new points into the t-SNE map.

Given the popularity of t-SNE, there is a surprising lack of research into the intricacies of running the method. The authors examine several of the algorithm's parameters and propose informed heuristics for setting them. This is an important step for researchers who utilize t-SNE in their work.

My comments of the manuscript follow.

- Overall, the structure of the manuscript is confusing. Results include text that should go to methods, the discussion is more of an extension of the results than an actual discussion, and the methods are organized poorly. This made reading the text more time-consuming and I suspect might alienate future readers.

Introduction

- The introduction makes no mention of other dimensionality reduction methods. While these are mentioned in the discussion, I think that the authors should include a paragraph here that provides a high-level view of PCA, UMAP, LargeVis, HSNE, and potentially force-based layouts of graph.

- The description of the perplexity parameter is overly informal. While van der Maaten's website does use the term "number of effective nearest neighbors", I think that the authors should follow a more formal definition (and follow it with the informal one). For example, "The perplexity is an entropy-based limit on the variance of the Gaussian which is used to smooth each of the high-dimensional data points. Effectively, it is a smooth measure of the number of nearest neighbors."

- The authors use the "PCA" acronym without including the full name of the method (principal component analysis).

- Box 1 is one of the nicest formats I've seen for the t-SNE equations (and much better organized than the 2008 paper). Well done!

Results

- The chief issue with the results is the lack of quantification. The authors lay out a compelling case for visualization but do not include any data to support their observations.

* For dimensionality reduction, the authors might want to examine a metric such as the distribution or mean of the ratio of low-dimensional to high-dimensional distance, percent preserved k-nearest neighbors, or percent neighbors that belong to correct cell subset label.

* For new cell embedding, the authors can run t-SNE on all N cells, then remove each cell, embed it, and calculate the distance between the embedding coordinates and the original coordinates.

- Additionally, I am surprised at the lack of flow or mass cytometry data. This is one of the most popular use cases for t-SNE in computational biology. Additionally, using cytometry data will offer larger data sets (much larger than the 1.3m cells used here) and provide stronger validation

opportunities (several annotated cytometry data sets were published).

- The Data Sets section can be moved to the methods. I suggest organizing it as a table rather than a list. The methods section itself includes several repeating pieces of information for each data set, these can go in the table as well. It will save space and make the data sets information more accessible.

- Similarly, pre-processing can be moved to the beginning of the methods (which includes a detailed explanation of each step). Figure 1 is unnecessary, this is standard procedure for scRNAseq data.

- "Inspecting initial t-SNE results often allows to discover small outlying clusters of cell doublets or other experimental artefacts", this is a problematic paragraph since 1) it has no backing in the paper and 2) it might encourage inexperienced readers to discard real rare subsets. Since the data used by the authors is already clean I recommend removing it.

Discussion

- The comparison to UMAP is excellent and deserves to be part of the results section. By applying the quantification methods mentioned above to both t-SNE and UMAP the authors will be able to better delineate the performance comparison.

- The discussion is too brief at one paragraph. I suggest that the authors extend it by listing some of the conceptual limitations of t-SNE and other methods that their approach addresses, potential extensions of the method, potential applications outside of those presented in the paper, pitfalls that users of the method should be aware of, future directions, or some combination of the above.

Methods

- Which computers were used for the analysis? There is some mention of potential specifications, but it's unclear what was used for this analysis.

Signed,
El-ad David Amir

The art of using t-SNE for single-cell transcriptomics.

Response to comments

We thank the reviewers for careful reading of our manuscript and for the thoughtful suggestions. We have substantially revised the manuscript in order to address the main concerns, and believe that it strongly improved. Very briefly, the most important changes are the following:

1. We added a thorough quantitative evaluation of the proposed pipeline;
2. We added a synthetic dataset to demonstrate the main ingredients and the main benefits of our procedure;
3. We performed parameter space testing;
4. We streamlined our procedure and now use the same parameters for all data sets;
5. We added the new largest scRNA-seq data set: Cao et al., Nature 2019, with over two million cells. This analysis nicely demonstrate the usefulness of our procedure;
6. We learned about a parallel submission to *Nature Communications* (Belkina et al.) and adapted our procedure to increase the synergy between the two papers.

Detailed replies to individual points are provided below.

Reviewer #1

The manuscript titled "The art of using t-SNE for single-cell transcriptomics" presents a protocol for using t-SNE for single-cell transcriptomics. It is well written and easy to follow. The authors propose a combination of preprocessing and initial dimensionality reduction of the data with PCA followed different custom initializations (by means of PCA or a t-SNE with fewer points, which can also be used to align datasets) with multiple perplexity values for the actual t-SNE computation to conserve global distances. While the results in many cases look very promising and the protocol might be a useful addition to the tool chain for analyzing single-cell transcriptomics, there is little to no formal evaluation and it seems quite ad-hoc. Arguably these techniques are hard to evaluate since we do have little understanding of the contents of given complex, high-dimensional data. However, we must be rigorous in the evaluation of our tools. I would propose to at least add a synthetic dataset that is well understood and do proper parameter space testing and quantitative evaluation of the results for the shown data.

These are good points, and we agree with the Reviewer that in our initial submission we relied on a rather informal and qualitative “expert” evaluation. These comments made us re-consider our strategy, and we have now implemented all three suggestions mentioned by the Reviewer.

The new Figure 1 presents a synthetic dataset that we designed to capture some crucial aspects of the scRNA-seq datasets (it consists of several Gaussian clusters in a hierarchical arrangement with varying degree of separation). It shows in what respects the default t-SNE is inadequate and demonstrates how our proposed steps can mitigate that.

We introduce three quantitative evaluation metrics and use them right in Figure 1. One metric is the average fraction of K nearest neighbours (we use $K = 10$) in the high-dimensional space that remain

within K nearest neighbours in the embedding. This quantifies preservation of the local, “microscopic” structure. Another metric is the same measure but computed for cluster centroids: the fraction of K nearest clusters in the high-dimensional space that remain within K nearest clusters in the embedding. This quantifies preservation of the “mesoscopic” structure. Finally, the third metric is the correlation coefficient between pairwise distances in the high-dimensional space and the pairwise distances in the low-dimensional space. This tends to quantify the global, “macroscopic” structure.

After discussing the toy example, we turn to the Tasic et al. 2018 data set (Figure 2) and show that our procedure leads to a marked improvement regarding the meso- and macroscopic metrics compared to the default t-SNE. Additionally, as requested by the reviewer, we scan the ranges of several parameters in Figure 3 and show that our pipeline chooses a good compromise between the different measures.

Main concerns:

I think the title does not fit the content of the paper. Besides strongly disagreeing with the phrase art (see below), after reading the title I expected and hoped for an explanation on how to use and interpret t-SNE embeddings that seems to be urgently needed by the community. The main text then, however, describes yet another protocol incorporating some preprocessing and (seemingly) ad-hoc parameterization.

In fact we think that our paper *is* about “how to use and interpret t-SNE embeddings”. Our main focus is on how to achieve better t-SNE embeddings, but substantial parts of the manuscript are about “how to use” t-SNE embeddings in actual research: e.g. projection of new cells onto an existing embedding or aligned embeddings of independent datasets — both suggestions we consider useful for everyday work with scRNA-seq data.

We discuss the word “art” below. Overall we continue to think that our title is adequate, but can discuss some alternatives if the Reviewer insists.

We address the issue of our suggestions being “ad-hoc” below.

While superficially, the visualizations presented in the results mostly look nice, there is no quantitative evaluation and even the qualitative evaluation is very superficial. Several times, the authors make strong statements that one or the other embedding is better with very little and vague to no justification.

As already mentioned above, we implemented several quantitative evaluation metrics together with a thorough analysis of a synthetic data set and the Tasic et al. dataset to address this concern.

It is worth stressing that the question of how to quantify the quality of any given embedding is an open and complicated research problem. While the metrics that we implemented provide useful quantifications, they are not a gold standard and so we believe that there still is room for and value in an “expert judgment”. We strengthen this point by discussing in more depth the advantages of our visualization of the 10x-dataset in Figure 8 and analysing an additional large scale data set with >2 Mio cells (Figure 9), which also show a clear advantage of our procedure.

In addition, we critically reviewed all our judgements and re-wrote the corresponding passages so that they always provide explicit justification.

E.g. p6 "In fact, perplexity 500 yields a surprisingly good visualisation", "The result, when initialised with PCA, is arguably the best t-SNE visualisation of this data set that we were able to obtain." p7 "Our t-SNE visualisation (Figure 4c) did not use any of that but was nevertheless at least as good as the original one." While the authors do sometimes point out some reasons why they deem one or the other result 'better' it is very hand-wavy and unstructured.

We rewrote these (and similar) passages. For the Tasic et al. 2018 data set to which the first two quoted sentences refer, we now base our comparisons on the numerical evaluations (see new Figure 2). In addition, we now use the same procedure throughout the manuscript.

In some cases I also do not agree with the assessment. For example let's consider Figure 5. To me, 5e seems like a rather good result (considering that plotting 1M points in a small rectangle is bound to result in dense maps). I see a lot of structure, why is issue #2 worse here than in the others, there is clear separation between clusters?

We believe the Reviewer meant Figure 7. There are two main problems with panel E (“default t-SNE” apart from the learning rate increased to 1000; this is default e.g. in `scanpy`). One problem is that some of the clusters are “torn apart” (e.g. cluster 8). This is the same problem as identified in the parallel work by Belkina et al. Another problem is that the global structure is largely destroyed: e.g. inhibitory clusters (26, 4, 13, 8, 24, 19) are not located close to each other but are scatter across the embedding.

We appreciate that this was not presented sufficiently clearly. We revised the text and extended the next figure, Figure 8, to have gene overlays for our preferred t-SNE (Figure 7B) as well as for the default t-SNE (Figure 7E). This clearly shows that our procedure does much better in preserving global structure (e.g. inhibitory neurons are mapped to a similar region of the space). We believe that this new part of Figure 8 makes the problems with the default t-SNE visually very clear.

Furthermore, overlaying the cluster colors gives a false impression when judging the quality. This information is not there when exploring a new dataset so we could only judge by structure or overlaying genes as in Figure 8.

We agree with the reviewer that Fig. 8 provides the “true” evaluation for the quality of the different tSNE maps in Fig. 7. That said, we find that in practice, almost every single new paper using scRNA-seq performs clustering and shows a t-SNE figure with overlaid clusters. We follow this tradition in our manuscript.

Of course the result of unsupervised clustering cannot provide an independent validation, but at least one expects to see a rough agreement between the clustering results and the t-SNE visualisation (e.g. that individual clusters are mostly continuous in the embedding). In fact, this is not the case for the default tSNE embedding shown in Fig. 7, as discussed above. Moreover, in actual practice, researchers usually provide a biological interpretation for each cluster (e.g. using known marker genes or other approaches). Once this biological interpretation is available, the arrangement of clusters in any embedding can be judged as being more or less biologically meaningful.

In general there should be a much more structured analysis of these results. What are the metrics to judge the quality of an embedding? How do we know the structure claimed to be better is not imposed by some of the preprocessing? Especially the initialization has great impact on the final embedding. But how do we know that it does not enforce structure that is not there?

Following this suggestion, we have introduced several metrics, as explained above.

Regarding initialisation: the structure that gets “enforced” by PCA initialisation is the structure of the first two PCs, which are by definition the two axes capturing the most variation in the data. While one might be able to come up with an example data set where this structure is not the most interesting, it cannot be a structure “that is not there”.

Another crucial consideration here, is that we use PCA initialisation scaled down to having standard deviation 0.0001 which is the same standard deviation as used for random Gaussian initialisation. In tSNE implementations that are currently used much of the visible structure depends on the random seed of the random number generator – we would rather think that this is “enforcing structure that is not there”.

We have revised the text to be clearer about these matters (see Results around the new Figure 2e) and explicitly discuss these issues in the discussion.

Apart from these theoretical arguments, our quantitative evaluations we added with the revision consistently support that PCA initialisation outperforms random ones (Figures 1–3).

The last point brings me to another concern. There is no formal evaluation of which structure is actually in the data. Then how can we know that structure in the t-SNE embedding is actually coming from the data and which is coming from the initialization. Especially initializing with a random sample has a danger of placing ‘unseen’ types in a structure they don’t belong. And these unseen types are supposedly of major interest.

If we understood the Reviewer correctly, then this comment concerns our approach to embedding very large data sets (Figures 7–9) where we begin with creating an embedding for of a random subset. We used $n = 25000$ subsets, which is only 1–2% of the large datasets with 1–2 mln cells overall. Indeed, it

is possible that some rare cell type will not be sufficiently represented in a random subset, and so will be inadequately initialised in the full t-SNE.

Here again it is crucial that the initialisation is scaled to standard deviation 0.0001, exactly as the random Gaussian initialisation in standard tSNE implementations. The default approach is to use random initialization where it is practically guaranteed that this rare cell type is not adequately initialized. We think that using a downsampling-based initialization can only be better.

** How is this tackled especially when downsampling to create the initial t-SNE? The authors show an example (of 10 cells?) in Figure 5, some of these seemingly show very large uncertainty/error, even excluding the biggest outliers. Also basically all of these examples are in the same major area of the embedding what happened in the other areas? Again, a quantitative evaluation is necessary.*

All cells in the Cadwell et al. 2016 data set are inhibitory neurons from layer 1, so it can be expected that they correspond to a small subset of the Tasic et al. 2018 data set. Indeed, we find that they are positioned mostly in the *Lamp5* area. For our Figure 5 we only show the *Vip/Lamp5* part of the embedding from Figure 2f. No Cadwell et al. cell was projected elsewhere. We made it clearer in the text and figure caption.

Following the suggestion of the reviewers, we now additionally quantified the positioning precision of the cells from the Tasic et al. data set (in a cross-validation spirit, Figure 5b). We find that the errors are relatively small: cells mostly end up within the same cluster.

** What downsampling approach is used? Random? What effect does this have on rare types that might get lost during downsampling?*

Indeed, for downsampling large data sets we use a random selection of cells. Alternatively, if cluster labels are available, one could use a stratified approach that would ensure that some cells from all clusters get selected.

** When used for point placement only, again, what happens with 'unseen' types when further optimization, as proposed by Berman et al (2014), is omitted?*

We thank the reviewer for bringing this up. As we wrote in the paper, this positioning method “assumes that for each new cell there are cells of the same type in the reference data set. Cells that do not have a good match in the reference data can end up positioned in a misleading way.” So this method should be used with caution. Note, however, that the additional optimization approach (as in Berman et al; see our revised manuscript for additional citations) would not help in this situation. Even with this additional optimization, each new cell will be attracted to its closest neighbours from the existing embedding. Unseen types will end up in an arbitrary and possibly misleading position. We revised our text to be clearer about this.

The additional optimisation has been implemented in the Python re-implementation of FIt-SNE, <https://github.com/pavlin-policar/openTSNE>. We found that it has only a small effect for the datasets in our manuscript, but the Python implementation `openTSNE` makes it very easy to use in case one prefers the approach with optimisation. We now mention this possibility in our manuscript.

Finally, in basically all of the steps of the protocol several parameters are needed, from the number of PCA dimensions to a plethora of parameters for the t-SNE itself. There is little justification for any of the used values except for that they are the standard values, or that the result was good and no analysis how changing the parameters impacts the results. I would like to refer the authors to Belkina et al. <https://doi.org/10.1101/451690> as an example of rigorous parameter analysis. Furthermore, for all of the presented datasets these parameters/and sometimes initialization method are different. This is not helpful. For the protocol to be useful in practice in the best case we would have robust parameters that do not need much tuning, but at least we need guidelines on how to adjust these parameters to a given dataset.

We thank the reviewer for bringing this up. First, we consolidated our protocol and now use the same parameters for all data sets throughout the manuscript. Some of these parameters scale with the sample size N : following Belkina et al. (which turned out to be a parallel submission to *Nature Communications* and we now took efforts to make sure that our papers reinforce each other), we now use $N/12$ as the learning rate. For the multiscale perplexity combination, we now always use a combination of perplexity 30 (default in most t-SNE implementations) with $N/100$, which is 1% of the sample size. Second, following the Reviewer’s advice, we performed extensive parameter space testing (new Figure 3). We find that our chosen parameter set provides a good compromise between the different quality measures and outperforms standard t-SNE both on synthetic and real data sets.

Other:

In my opinion the term art and the authors claim that "In data visualization, as a matter of principle, there is just no "one size fits all" solution — the reason we call this "the art of using t-SNE"." are strongly misleading. While indeed there are rarely one size fits all solutions in visualization (or other computational sciences for that matter) a proper visualization (-method) is a result of rigorous analysis and a solid theoretical foundation and justification. As laid out above, most of this is missing in the paper.

We removed the quoted sentence from the Discussion. In the revised version of the manuscript we are using the same protocol for all data sets, hence we do not need the discussion about the absence of “one size fits all” solution. Our protocol (and t-SNE itself) certainly has limitations and they are more explicitly discussed in the Discussion.

That said, we prefer to keep the title as it is. We understand “the art of using t-SNE” in the same sense as it used in Knuth’s “The Art of Computer Programming”: t-SNE is a complicated tool and the point of the paper is to explain how to use it effectively in this particular applied field. We can discuss alternative titles if the Reviewer insists.

Some computational performance numbers would be helpful. Some of the proposed techniques seem quite expensive (large perplexity, nearest neighbor search for point placement (I assume hence the $k=1$ for large datasets), etc.)

We have inserted run times throughout the text. To clarify, $K = 1$ for large datasets was done mainly for simplicity of implementation. This initial positioning runs in several minutes and is not a bottleneck.

p4. "In general, inspecting initial t-SNE results often allows to discover small outlying clusters of cell doublets or other experimental artefacts. We recommend discarding such cells and re-running the analysis: otherwise such artefact clusters can create "bridges" between unrelated t-SNE clusters and spoil the visualisation." How do we know those bridges are artifacts and not signal. And if they could be signal aren't those bridges actually helpful to keep global structure?

The other Reviewer suggested to remove this paragraph and we now followed this advice. We are not doing any data cleaning and are not discarding any cell doublets in this paper: all data sets are used here as they were used in the original publications.

p5. "we will, from now on, not use cluster labels for any analysis." This is a bit confusing. I assume the authors mean they do not use the cluster labels as input for the t-SNE (they use them for the visuals in basically all figures). Does this mean for Figure 2 they used the labels as input?

Yes, we show MDS of cluster centroids in the Figure 2, and one needs cluster labels in order to compute cluster centroids. What we meant here was indeed that cluster labels were never an input for t-SNE. We reformulated for clarity.

p6. "perplexity 500 yields a surprisingly good visualisation." Is this really surprising? Isn't that exactly what we know a larger perplexity does?

This is a fair comment. We have reformulated this sentence. “Suprisingly” originally referred to the fact that so large perplexities have never (to the best of our knowledge) been used for scRNA-seq data analysis, so it might come as a surprise that they can in fact be useful.

p11. "Validating this procedure is not straightforward, as there are no annotations available for the clusters." Weren't these used in Figure 7 for coloring?

What we meant is that while there are cluster labels for this data set (that we took from the Wolf et al. paper on `scanpy`), they are not annotated: for many Wolf et al. clusters it is unknown what biological cell types they represent. In fact, this is the only data set among the ones we use that was released (by 10x Genomics) without an associated paper.

That said, we have reformulated/removed this sentence. We now do provide a validation – independent of cluster identify – in Figure 8.

Reviewer #2

The authors investigated the various parameters of the t-distributed Stochastic Neighbor Embedding (t-SNE) algorithm in the context of several scRNAseq data sets, the largest of which included 1.3 million points. They integrate two existing heuristics, PCA initialization and multi-scale combination of perplexity values, and show that these lead to improved t-SNE results. Additionally, the authors show that k-nearest neighbors classification can be used to embed new points into the t-SNE map.

Given the popularity of t-SNE, there is a surprising lack of research into the intricacies of running the method. The authors examine several of the algorithm's parameters and propose informed heuristics for setting them. This is an important step for researchers who utilize t-SNE in their work.

We thank the Reviewer for the positive assessment.

My comments of the manuscript follow.

- Overall, the structure of the manuscript is confusing. Results include text that should go to methods, the discussion is more of an extension of the results than an actual discussion, and the methods are organized poorly. This made reading the text more time-consuming and I suspect might alienate future readers.

We made an effort to organize the text better, following the Reviewer's specific suggestions below.

Introduction

- The introduction makes no mention of other dimensionality reduction methods. While these are mentioned in the discussion, I think that the authors should include a paragraph here that provides a high-level view of PCA, UMAP, LargeVis, HSNE, and potentially force-based layouts of graph.

We inserted a reference to UMAP into the Introduction, but the other methods listed here are (to the best of our knowledge) not being used in the scRNA-seq community, so we feel a discussion of them can be postponed until the Discussion.

- The description of the perplexity parameter is overly informal. While van der Maaten's website does use the term "number of effective nearest neighbors", I think that the authors should follow a more formal definition (and follow it with the informal one). For example, "The perplexity is an entropy-based limit on the variance of the Gaussian which is used to smooth each of the high-dimensional data points. Effectively, it is a smooth measure of the number of nearest neighbors."

We edited the text to elaborate on the meaning of perplexity.

- *The authors use the “PCA” acronym without including the full name of the method (principal component analysis).*

Fixed.

- *Box 1 is one of the nicest formats I've seen for the t-SNE equations (and much better organized than the 2008 paper). Well done!*

Thank you!

Results

- *The chief issue with the results is the lack of quantification. The authors lay out a compelling case for visualization but do not include any data to support their observations.*

We thank the reviewer for bringing this up. It was also one of the major critiques by the other Reviewer. We have included additional analysis using several measures to quantify different aspects of the quality of an embedding and use them to support our conclusions. To copy our response to Reviewer 1:

We introduce three quantitative evaluation metrics and use them right in Figure 1. One metric is the average fraction of K nearest neighbours (we use $K = 10$) in the high-dimensional space that remain within K nearest neighbours in the embedding. This quantifies preservation of the local, “microscopic” structure. Another metric is the same measure but computed for cluster centroids: the fraction of K nearest clusters in the high-dimensional space that remain within K nearest clusters in the embedding. This quantifies preservation of the “mesoscopic” structure. Finally, the third metric is the correlation coefficient between pairwise distances in the high-dimensional space and the pairwise distances in the low-dimensional space. This tends to quantify the global, “macroscopic” structure.

** For dimensionality reduction, the authors might want to examine a metric such as the distribution or mean of the ratio of low-dimensional to high-dimensional distance, percent preserved k -nearest neighbors, or percent neighbors that belong to correct cell subset label.*

We thank the Reviewer for these suggestions. As we wrote above, we now use a fraction of preserved K -nearest neighbours as one of the metrics.

** For new cell embedding, the authors can run t-SNE on all N cells, then remove each cell, embed it, and calculate the distance between the embedding coordinates and the original coordinates.*

We thank the Reviewer for this suggestion. We performed this analysis and found that most cells are positioned in their original cluster (see figure 5b and the surrounding text).

- *Additionally, I am surprised at the lack of flow or mass cytometry data. This is one of the most popular use cases for t-SNE in computational biology. Additionally, using cytometry data will offer larger data sets (much larger than the 1.3m cells used here) and provide stronger validation opportunities (several annotated cytometry data sets were published).*

We thank the reviewer for this suggestion. We have considered it but decided against including non-transcriptomic data sets. We are not experts in cytometry and our manuscript is focused entirely on scRNA-seq (it is even in the title!). During the revision we have learnt that there is a parallel submission to *Nature Communications* by Belkina et al. who discuss t-SNE in the context of cytometric data sets. We think there is a good synergy between our papers and it makes sense that we focus on scRNA-seq while they focus on cytometry.

- *The Data Sets section can be moved to the methods. I suggest organizing it as a table rather than a list. The methods section itself includes several repeating pieces of information for each data set, these can go in the table as well. It will save space and make the data sets information more accessible.*

We followed these suggestions.

- *Similarly, pre-processing can be moved to the beginning of the methods (which includes a detailed explanation of each step). Figure 1 is unnecessary, this is standard procedure for scRNAseq data.*

We followed these suggestions and removed Figure 1.

- *“Inspecting initial t-SNE results often allows to discover small outlying clusters of cell doublets or other experimental artefacts”, this is a problematic paragraph since 1) it has no backing in the paper and 2) it might encourage inexperienced readers to discard real rare subsets. Since the data used by the authors is already clean I recommend removing it.*

We followed the advice and removed the sentence.

Discussion

- *The comparison to UMAP is excellent and deserves to be part of the results section. By applying the quantification methods mentioned above to both t-SNE and UMAP the authors will be able to better delineate the performance comparison.*

We extended our comparison to UMAP and included two supplementary figures. That said, we are *not* claiming that our t-SNE pipeline outperforms UMAP as our manuscript is not supposed to be a comparison of t-SNE with other methods. What we do want to say is that when t-SNE is run appropriately, then UMAP is not as a clear winner as it is sometimes being presented in the literature (e.g. UMAP preprint itself; or Becht et al. 2018, Nature Biotechnology).

- *The discussion is too brief at one paragraph. I suggest that the authors extend it by listing some of the conceptual limitations of t-SNE and other methods that their approach addresses, potential extensions of the method, potential applications outside of those presented in the paper, pitfalls that users of the method should be aware of, future directions, or some combination of the above.*

These are excellent suggestions. We very substantially revised the Discussion and covered many of the topics listed here.

Methods

- *Which computers were used for the analysis? There is some mention of potential specifications, but it's unclear what was used for this analysis.*

We edited to clarify.

REVIEWERS' COMMENTS:

Reviewer #2 (Remarks to the Author):

The authors have followed the suggestions from both reviewers faithfully and greatly expanded the scope and utility of the manuscript. I have no further comments.

-- El-ad David Amir